# Antiviral Peptides in Antimicrobial Surface Coatings—From Current Techniques to Potential Applications

**DOI:** 10.3390/v15030640

**Published:** 2023-02-27

**Authors:** Mahe Jabeen, Payel Biswas, Md Touhidul Islam, Rajesh Paul

**Affiliations:** 1Department of Chemical and Biomolecular Engineering, North Carolina State University, Raleigh, NC 27606, USA; 2Department of Chemical Engineering, University of Rhode Island, Kingston, RI 02881, USA; 3Department of Civil, Environmental, and Construction Engineering, University of Central Florida, Orlando, FL 32816, USA

**Keywords:** peptides, antimicrobial coatings, surface coatings, antimicrobial peptides, antiviral peptides

## Abstract

The transmission of pathogens through contact with contaminated surfaces is an important route for the spread of infections. The recent outbreak of COVID-19 highlights the necessity to attenuate surface-mediated transmission. Currently, the disinfection and sanitization of surfaces are commonly performed in this regard. However, there are some disadvantages associated with these practices, including the development of antibiotic resistance, viral mutation, etc.; hence, a better strategy is necessary. In recent years, peptides have been studied to be utilized as a potential alternative. They are part of the host immune defense and have many potential in vivo applications in drug delivery, diagnostics, immunomodulation, etc. Additionally, the ability of peptides to interact with different molecules and membrane surfaces of microorganisms has made it possible to exploit them in ex vivo applications such as antimicrobial (antibacterial and antiviral) coatings. Although antibacterial peptide coatings have been studied extensively and proven to be effective, antiviral coatings are a more recent development. Therefore, this study aims to highlight antiviral coating strategies and the current practices and application of antiviral coating materials in personal protective equipment, healthcare devices, and textiles and surfaces in public settings. Here, we have presented a review on potential techniques to incorporate peptides in current surface coating strategies that will serve as a guide for developing cost-effective, sustainable and coherent antiviral surface coatings. We further our discussion to highlight some challenges of using peptides as a surface coating material and to examine future perspectives.

## 1. Introduction

In light of the continuing global pandemic caused by COVID-19, there is a greater focus on implementing the safest possible procedures to prevent the transmission of viruses. The COVID-19 pandemic affected people on a global scale and had lasting repercussions for individuals, communities, and societies in terms of health, economics, society, culture, etc. Therefore, preventing the transmission of viruses is essential for protecting human health, easing the burden on healthcare systems, and maintaining economic stability. Viruses are self-contained biological organisms consisting of a DNA or RNA core surrounded by a protein shell [1]. Viruses can be classified by their nucleic acid (RNA or DNA) [2], lipid membrane [3], shape [4], etc. Lipid membrane classification is the most common, and they can be classified as either enveloped, meaning that their viral particle is surrounded by a lipid membrane, or non-enveloped. Viruses cannot multiply until they infect a host cell [5]. Envelopes are often formed from the plasma membrane of the host cell during budding, the process by which viruses leave their host cell [6]. The envelope is then modified by the addition of proteins (glycoproteins) in the form of spikes, which aid in the virus’s entry into host cells and, in conjunction with the envelope, play a variety of roles in virus–host interactions [7]. This mechanism is demonstrated in Figure 1. While the envelope is important for the process of budding from the cell, it also facilitates structural flexibility and serves to mask capsid spike antigens from antibodies produced by the host [8]. Non-enveloped viruses can invade cells via a number of endocytic pathways that ultimately result in plasma membrane penetration, or via the internal penetration of membranes [9]. This capability to avoid the host’s immune response could be a key factor in the spread of viral infections.

Viruses are capable of rapid mutation when they enter and proliferate within their host. The virus is transferred and propagated among humans by various routes, including air, direct contact, body fluid transmission, and indirectly through contaminated shared surfaces. In addition, many viruses are transmitted to humans by animals or insects. Respiratory viruses such as influenza [11] and SARS-CoV [12] are spread through droplets expelled by an infected host into the air. As observed with SARS-CoV-2, this mode of respiratory virus transmission is fast and difficult to contain [13]. This transmission is shown in Figure 2. Unlike viruses, when bacteria infect a human they tend to stay in one region and spread from there, resulting in what is called a “local infection” [1,14]. Because of this, it is far simpler to treat bacterial infections with novel antimicrobials than it is to treat viral infections. Due to the rapid mutation of the virus’s structure, the creation of viral vaccines is time-consuming and associated with failure risk. In addition to creating conventional vaccinations, novel materials and coatings with broad potency against multiple microorganisms are necessary.

It is well-established that contaminated surfaces are a significant factor in the dissemination of viral diseases [16,17]. Surface contact can spread viruses such as influenza, Hepatitis B, respiratory syncytial virus (RSV), rhinoviruses, noroviruses, and coronaviruses [18,19]. Because of their ability to cause severe disease and other health problems in humans, viruses have long been considered a growing threat to society [20]. For example, the coronavirus disease (COVID-19) has emerged as a catastrophic threat to global human health [21]. A recent outbreak of Ebola hemorrhagic fever (also known as EHF) (2014) has had a devastating impact on the living species in Africa [22,23]. In recent years, there has been rising concern over the emergence of novel, more dangerous viruses, especially with SARS [24]. This has rekindled interest in the quest to develop surfaces (antimicrobial surfaces) that can inhibit the propagation of viruses and other microorganisms. While several antibacterial coatings have been developed and put into commercial use, relatively little is known about antiviral coatings [25]. Antiviral surface studies are often restricted by the complexity of viral structures, various varieties of viruses, and a lack of molecular understanding of these non-living particles.

Depending on the conditions, respiratory viruses can live on inanimate environmental surfaces for a long time [17]. The prevalence and spread of infectious viruses can be mitigated through effective cleaning and disinfection methods [26]. There are many different types of antimicrobial coatings, but they all serve the same purpose: to prevent the growth of viruses and other microorganisms on the treated surface. It is important that any antimicrobial coating be able to cover any surface regardless of the applied environmental conditions mechanically, and that it be able to inactivate any virus or microorganism quickly without leaving detrimental consequences for the consumers [27]. Many kinds of substances have been considered for use as antimicrobial coatings. Each option comes with its own set of benefits and drawbacks. For example, metal nanoparticles such as silver nanoparticles are a type of antimicrobial agent normally immobilized or adsorbed onto the target surface. However, silver nanoparticles could be harmful to other life forms. Low-dose exposure to silver nanoparticles causes oxidative stress and mitochondrial dysfunction, according to an in vitro toxicity assessment in rat liver cells [28]. Additionally, silver nanoparticles were found to be harmful to mouse germline stem cells in vitro, where they disrupted mitochondrial function and induced cell membrane leakage [29,30,31]. Copper, on the other hand, may be easily incorporated, e.g., as an alloy or coating, into commonly touched hard surfaces such as door accessories, faucets, stair banisters, and steadying poles in transportation [32]. However, the potential adverse effects of relatively high copper usage must be carefully considered. Neurodegenerative diseases such as Alzheimer’s and Parkinson’s may have a connection to copper toxicity, or copper may contribute to the development of these diseases [33]. One of the most important factors to consider while deploying antimicrobial surfaces is the potential for unintended environmental impacts due to leaching [34]. Nature can serve as a source of inspiration for the development of antiviral techniques, and biopolymers originating from natural sources may provide a promising avenue, similar to the discovery of naturally occurring antibiotics such as antimicrobial peptides (AMPs) [5]. The skin acts as a physical barrier to protect against the outside environment and as the body’s first line of defense against pathogenic microorganisms such as bacteria, viruses, and fungi. The epithelial surface of the skin contains the cellular and performed biochemical components that make up the innate immune system. In the skin, soluble peptides known as antimicrobial peptides (AMPs) play a crucial role in the innate immune system’s defense against pathogens [35]. In this review, the peptides that have potency against viruses will be termed antiviral peptides (AVP).

Antiviral peptides (AVPs) are short chains of amino acids that have the ability to inhibit viral infections by acting on different stages of the virus life cycle. AVPs belong to the broad class of antimicrobial peptides (AMPs), which are part of the immune system of all living organisms. AVPs are gaining prominence as novel therapeutic targets, as the peptides have antiviral efficacy to inhibit viral infection directly and indirectly. Studies suggest that AVPs have the capacity to target various steps in the viral life cycle, from attachment to the host cells to the viral replication system in the host. However, the mechanism of action and target inhibition sites of these AVPs in the viral replication cycle varies depending on the type of peptide and the viral pathogen. AVPs can be obtained from either natural, synthetic, or recombinant sources. Synthetic AVPs are created by artificially adding chemical groups or amino acids to the naturally occurring peptide sequences. As with AMPs, the naturally occurring AVPs have a net positive charge and are cationic and amphipathic in nature. These AVPs can come from various sources, including plants, bacteria, arthropods, amphibians, marine animals, and mammals, with a wide range of action mechanisms.

Due to low toxicity, high specificity, and negligible side effects, AVPs have become attractive novel therapeutic options. The surface material, its characteristics, and the environment around it are all important contributors to long-term viral persistence. The type of surface, its porosity, and its adsorption sites can all have an impact on how long viruses remain on surfaces. Physical factors such as temperature, humidity, and surface roughness also contribute to virus persistence. Viral persistence may also be facilitated by biological variables, including the presence of other bacteria, biofilms, or biological fluids such as saliva or mucus droplets. Lastly, chemical factors such as pH, reactive species, or the presence of organic materials can also make it worse as it has been shown that many viruses can be stabilized by the presence of organic materials such as lipids and proteins in the environment [36]. It is important to tailor the surface coatings specifically for the virus type because each type interacts with the surface differently. Material properties, along with environmental factors, should be considered for designing efficient antiviral coatings. This literature review focuses on peptides, which are natural substances, and their antimicrobial properties, primarily to bring attention to the antiviral peptide as a novel material in coating strategies. The current coating techniques and the potential of antiviral peptides in surface coatings are discussed in Section 2 and Section 4, respectively. In addition, the potential challenges of working with antiviral peptides and their future prospects are described.

## 2. Current Antiviral Surface Coating Techniques

Viruses spread mainly through aerosolized particles. Large droplets (>20 um), when they land on a surface, evaporate and spread through the air. The virus on the surface remains active for several hours [37]. In this way, the contaminated surface helps the virus to propagate from sick to healthy individuals. Viruses such as SARS-CoV-2 can remain active on surfaces for up to 6.5 h [38]. Therefore, inactivating viruses as soon as they land on the surface is the best possible way to mitigate their spread. Surfaces can be sterilized by using sanitizer or household cleaners. However, it is not a viable strategy to sterilize surfaces after each individual use, especially in places with a high density of infected patients, such as hospitals, senior living facilities, etc. Antiviral coatings have been in use for virus inactivation for decades and show promise to combat viral outbreaks [38]. The classification of current antiviral surface coating materials is illustrated by Figure 3 below, and a brief discussion for each class is presented thenceforth.

### 2.1. Metal Ions/Metal Oxide

Numerous studies have tested the virucidal effects of various metal ions, including copper, silver, gold, zinc, etc., and their oxides [40]. Viruses, unlike bacteria, do not have nucleic acid repair systems, making them more vulnerable to metals that cause nucleic acid damage. Metals and metal oxides have two antiviral modes of action, either metal reduction potential or metal donor atom selectivity. The metal reduction potential mechanism results in the production of reactive oxygen species (ROS) through redox reactions. ROS generation is supplemented by the formation of the covalent bond between metal atoms and sulfur. ROS induces oxidative stress inside the cell, leading to the disruption of cellular contents including viral genomes. The metal donor atom selectivity involves the binding of metal ions or atoms to the donor atoms, causing the disruption of organelles which leads to oxidative stress. Figure 4 illustrates two antiviral mechanisms of copper ions.

“In numerous literatures, It has been demonstrated that copper oxide exhibits antiviral properties against both enveloped and nonenveloped DNA and RNA viruses [40]”. Copper oxide converts between Cu (I) and Cu (II) oxidation states where Cu (II) ions can oxidize and damage the biomolecules. Copper particularly targets the genes necessary for viral infectivity in the viral genome [41]. It can disorder both types of DNA and RNA by adhering to them and cross-linking between their strands [42]. It has been found to be effective against the influenza A virus, avian influenza H5 subtype virus, and HIV-1 virus.

### 2.2. Nanoparticle-Based Materials

Metal-based nanoparticles such as Cu, Ag, Au, Zn, etc., are commonly used as antiviral agents due to their ability to generate ROS, leading to the physical disruption of the cell membrane. They inhibit the binding between the virus and host cell by interacting with the glycoproteins of the virus cell. They also bind with the viral factor and the cellular factor of the host cell, consequently blocking viral replication. Figure 5 illustrates the antiviral mechanism of nanoparticles.

Silver nanoparticles (AgNPs) have shown significant efficacy against bacteria, viruses, and even eukaryotic organisms [44]. They can take rapid and effective antiviral action against a wide range of viruses, including respiratory syncytial virus, norovirus, influenza virus, herpesvirus, hepatitis B virus, and human immunodeficiency virus [45]. They have been proven to be effective against SARS-CoV-1, SARS-CoV-2 [46], and human coronavirus HCoV-OC [47] when employed as a virucidal agent [48]. Anti-viral coatings made of SiO_2_/Ag composites can be used as a new method to fight against SARS-CoV-2 [49]. Antelman et al. found that silver as tetrasilver tetroxide (Ag_4_O_4_) is effective against HIV-1 at concentrations between 1 and 20 ppm [48].

Gold nanoparticles (AuNPs) suppress viral infection in a similar mechanism, where they block the virus membrane fusion to the host cell and prevent the virus entry into the host cell. Gold nanoparticles are effective in inactivating enveloped viruses and have shown promise to inhibit measles [5]. Metal nanoparticles can be incorporated into textiles, latex and polymers, clothing during manufacture, wound dressings, implant coatings, etc. Additionally, textiles with a boron–triclosan mixture have shown efficacy against the human adenovirus and poliovirus type 1 strain. However, exploiting metal nanoparticles in the antiviral coating has some side effects. Metal nanoparticles have cytotoxicity, especially at high concentrations. They can also trigger allergic reactions in some individuals. Another potential side effect is the development of viral resistance to the nanoparticles, making the coating less effective over time. Last but not least, metal nanoparticles have a negative environmental impact. They can end up in the water bodies and disrupt the aquatic ecosystem. Therefore, it is necessary to explore alternative antiviral agents with fewer side effects [50].

### 2.3. Carbon-Based Materials

Graphene is a promising carbon-based antimicrobial material currently used for antiviral surface coating. It is a two-dimensional sheet of sp2-hybridized carbon atoms organized in a hexagonal lattice. It has shown exceptional antimicrobial capabilities, including excellent electrical and thermal conductivity [51]. Their antimicrobial property is attributed to their reduced form as graphene nanoplatelets (GNP) and reduced graphene oxide (rGO), as well as their oxidized form as graphene oxide (GO) [52]. Multiple viruses, including pseudorabies, tomato bushy stunt virus, respiratory syncytial virus, and herpes simplex virus, have been proven to be inhibited by GO’s antiviral properties [52,53,54,55]. Graphene nanoparticles can puncture the lipidic virus membrane through sharp knife-like edges. GO, rGO, and GNP materials are used as components of nanocomposites where metal nanostructures covalently attach to them [56]. Graphene has been proven to operate as an antiviral/virucidal agent and is used to produce antibacterial surfaces. However, no graphene surfaces have been tested yet for antiviral characteristics [57].

### 2.4. Synthetic Polymer-Based Materials

Synthetic polymers, sometimes known as “man-made” or “human-made” polymers, are produced through chemical reactions from monomers, which are tiny molecules that can be linked to form larger polymer chains. Polymers are now used for antimicrobial surface coating production due to their diverse macromolecular chemistry, which aids in modifying the physicochemical features of polymers [58]. The cationic polymer is a synthetic polymer that has gained attention to combat viral outbreaks. Electro-positively charged groups in the polymer chain make the cationic polymer the most popular antiviral agent. Cationic polymers kill viruses by a contact-dependent mechanism that does not involve chemical release (Figure 6) [59,60]. Additionally, it is resistant to the adhesion of viruses and microorganisms to its surface [61].

Hydrogel is a three-dimensional network of hydrophilic polymers that can absorb a considerable amount of water without losing its structure [62]. Polymers used to make hydrogel coatings can also have antiviral effects in particular circumstances [63]. For example, when tested against enveloped viruses such as VSV, HSV, visna virus, and HIV, monocarpin was found to be a highly potent antiviral monoglyceride [64].

UV-enhanced polycations are cationic synthetic polymers engineered to exhibit enhanced responsiveness to UV light. Cationic conjugated polyelectrolytes (CPE) and oligophenylene ethynylenes (OPE), derived from poly (phenylene ethynylene), have been demonstrated to exert considerable photoinducible antiviral activity. Subjecting the material to UV light exposure substantially increased its antiviral properties. In comparison to the dark studies, the virucidal activity was significantly increased [65,66,67].

### 2.5. Natural Polymer-Based Materials

There are many naturally occurring materials that are used as antiviral coatings. Chitosan is a common antimicrobial material derived from naturally occurring polymer chitin [39]. It is effective against human immunodeficiency virus (HIV), hepatitis B virus (HBV), and influenza virus. The antiviral mechanism involves physical inhibition, which blocks the entry of the virus cell into the host cell, the enzymatic degradation of the virus cell, interference with the viral replication, etc.

Lignin is another common plant-derived polymer that has shown promise to be used as a coating material. It has strong antiviral properties, which can be attributed to the local generation of reactive oxygen species (ROS) upon exposure to light. The ROS can react with the amino acid of the capsid or envelope and cause oxidative disruption of the virus cells. Boarino et al. showed that lignin is effective against Herpes Simplex Virus type 2 (HSV-2). Lignin can be used as a cost-effective and environmentally sustainable polymer-based antiviral and antimicrobial coating [68].

Cellulose is used in food products such as meat and as a wax coating on fruits and vegetables. However, it does not have intrinsic antiviral properties to kill viruses. It can be modified or combined with other materials to render antiviral properties. For example, when incorporated with copper nanoparticles, cellulose reduces the infectivity of SARS-CoV-2 [69].

## 3. Peptides and Antiviral Activity

Antimicrobial peptides (AMPs) are recognized as naturally occurring molecules that are components of the host defense and innate immune systems in all living species [70]. The discovery of AMPs was first made in plants, followed by amphibians, and became popular as host defense molecules in the 1980s during the emergence of multi-drug resistant pathogens. These peptides have now been discovered in microorganisms, mammals, fish, and invertebrates and identified in all species of life [70,71]. The Antimicrobial Peptide Database now has over 2500 AMPs. However, this list probably only includes a small portion of the naturally occurring, gene-encoded peptides. Antimicrobial peptides (AMPs) have been extensively studied in the past, with a focus primarily on their antibacterial activity and potential as an alternative to conventional antibiotics. These peptides also have a rapid response against other organisms such as viruses, bacteria, fungi, and protozoa [72]. Some AMPs have also been identified with anticancer properties against human lung and bladder cancer cells [70,73].

AMPs consist of typically short peptide chains (10–50 amino acid residues) with a net positive charge. These peptides are cationic and typically have hydrophobic residues to interact with cell membranes made of anionic phospholipids. The hydrophobicity of AMPs and electrostatic interaction with pathogen membranes are important properties in fighting against bacteria and enveloped viruses. Most antimicrobial peptides work by interacting with anionic membrane phospholipids, resulting in pore formation, disruption, the leakage of cell metabolites, and lysis of the target cell. Some AMPs can display more complex behaviors that involve targeting specific binding sites and intracellular pathways and have more than one antimicrobial mechanism [74]. However, only a small number of AMPs have been made available for use as potential drugs, and only a small number of AMPs make it to clinical trials [75]. The 36-amino acid residue peptide Enfuvirtide (Enf) was the first approved peptide drug for clinical use against human immunodeficiency virus (HIV), and blocked the fusion of the heptad repeat 1 (HR1) domain with the HR2 domain, inhibiting the infection. Peptide-based therapies have also been introduced against other viruses such as influenza, hepatitis, herpes, etc. Synthetic peptides, Boceprevir and Telaprevir against Hepatitis C virus (HCV), received approval from the FDA in 2011. Freitas et al. also summarized antiviral peptide-based drugs under preclinical or clinical trials against enveloped viruses [76].

Agarwal et al. discuss the validation and approval of different antiviral peptides and address peptide sources from natural, computational, and biological approaches [77]. Figure 7 refers to the current sources of antiviral peptides. Recombinant DNA technology is another source of peptides in addition to natural and synthetic sources (Figure 7b). Chemical processes are used to create synthetic peptides, which are small polymers of amino acids. The synthesis reaction entails attaching an amino acid’s carboxyl group to the amino group of the following amino acid in the peptide chain. In recombinant peptide production, a modified peptide encoded by recombinant DNA is cloned in a foreign expression system. Recombinant production gives more control over the peptide sequence, higher yields, and is a more sustainable source of the peptides [78]. However, this technique requires initial work on proper plasmid construction and genetic design. Additionally, the downstream purification process can be costly for bacterial expression vectors, as the peptides are not secreted in the media and bacterial endotoxins need to be removed before commercialization. After purification, these recombinant peptides have unimpaired antimicrobial properties in vitro and also have therapeutic potential [79,80,81].

There are several review articles on antimicrobial peptides derived from farm animals and venomous animals and their mode of action and application [82,83].

In this section, we have mostly concentrated on the antiviral peptides that are present in mammals, particularly in humans. In mammals, the two primary AVP families are defensins and cationic cathelicidin peptides, both of which have demonstrated efficacy against a variety of viruses. There are numerous defensin genes, but hCAP-18 is the only known cathelicidin gene in humans [82,83].

### 3.1. Human Cathelicidin and Defensin: Mechanism of Action against Pathogenic Viruses

#### 3.1.1. Human Cathelicidin

Cathelicidin, an important class of antiviral peptides, is evolutionarily conserved in many eukaryotes, from amphibians, fish, and reptiles to mammals. In humans, these peptides are mostly expressed on neutrophils and monocyte cells, which are part of the innate immunity system. These peptides are also expressed in keratinocytes on the skin and mucosal epithelium, as well as at the site of inflammation as a primary defense mechanism and a natural antibiotic in the host. Human cathelicidin, hCAP-18, does not have antimicrobial properties unless it is cleaved by proteinase 3 and active LL-37 peptide is generated [85].

The most prominent human cathelicidin peptide, LL-37, has shown a wide range of antimicrobial activity against bacteria, enveloped viruses, and fungi and has the ability to inhibit bacterial biofilm formation [86,87,88]. By activating immune cells, cathelicidin also modifies the adaptive immune system [89]. The knockout of cathelicidin in mice shows greater mortality due to higher susceptibility to bacterial and viral infections and proves the biological relevance of cathelicidin in the immune system. CRAMP (mouse homolog to LL-37)-deficient mice also showed significantly worse infections than control healthy mice [90]. The C-terminal end of LL-37 contributes to the wide range of antibacterial efficacy of this specific peptide. This C-terminal end also has bacterial lipopolysaccharide (LPS) binding activity. LPS is a bacterial endotoxin, and cathelicidin can neutralize these toxins and has promising applications in therapeutics [91]. Cathelicidin can inactivate both Gram-positive and Gram-negative bacteria through its antibacterial mechanisms. For Gram-negative bacteria, these peptides interact with both the inner and outer membranes. Cathelicidin interacts with lipopolysaccharides (LPS) on the bacterial surfaces and accumulates before entering the cells. After invading the cells through membrane disruption, those peptides prevent infection by killing bacteria. For Gram-positive bacteria, the mechanism is not well known. However, some Gram-positive bacteria show efficacy against cathelicidin at a low dosage [92].

Its mechanism of action against viruses may vary depending on the type of virus. Common mechanisms include the disruption of the viral membrane or the inhibition of viral attachment to host cell receptors, thus preventing entry into the host. LL-37 has shown antiviral potency against many viral pathogens such as Respiratory Syncytial Virus (RSV), dengue virus, Human Immunodeficiency Virus-1 (HIV-1), Herpes Simplex Virus (HSV), vaccinia virus and Adenovirus, influenza A virus, Rhinovirus (HRV), Zika virus, Ebola virus, Coronavirus, etc. [93,94].

#### 3.1.2. Human Defensin

Defensin peptides are also part of the innate immune system of the host. These peptides are expressed primarily in neutrophils and epithelial cells and are also conserved in plants, insects, mammals, and birds. Defensin peptides consist of six cysteine residues forming three disulfide bonds, and are classified into three subfamilies based on the arrangement of disulfide bonds: *α*-defensins, *β*-defensins, and *θ*-defensins. Human defensins have only two subfamilies of cationic peptides: *α*-defensins and *β*-defensins. *α*-defensins are referred to as human neutrophil peptides and are further divided into two classes according to their structure and expression pattern: human neutrophil peptides 1 to 4 (HNPs) and enteric human defensins 5 and 6 (HDs). HDs are constitutively expressed in small intestinal cells, whereas HNP secretion is regulated in the granules of human neutrophils. The expression of human *β*-defensins (HBD) is restricted to epithelial cells and keratinocytes. The expression of *β*-defensins is transcriptionally regulated. HBD1 is also constitutively expressed, like HDs. However, HBD2 and HBD3 are induced in response to infection and inflammation. These cationic peptides have wide antimicrobial activity against a variety of microorganisms, including bacteria, viruses, tumors, fungi, and even tumor cells. The primary factor in defensin’s antibacterial activity is how it interacts with the organism’s membrane or envelope due to its amphipathic nature [95].

Defensins can target multiple virus–host interaction steps and have antiviral activity against both enveloped and non-enveloped viruses. Defensins can prevent the adhesion of enveloped herpes simplex virus-2 (HSV) to the host cell by interfering with the receptor. Another mechanism of action is to prevent virion fusion to the cell membrane, as observed with the influenza virus and HIV-1. In non-enveloped viruses such as human adenovirus, defensins target intracellular inhibition by stabilizing the capsid protein and protecting host cells by preventing the interaction of viral machinery and host factors. Wilson et al. detail the antiviral action and the multitude of ways through which human defensin peptides suppress viruses [96]. However, *α*-defensins might have a contradictory role in viral infection enhancement [95].

In Figure 8, Mulder et al. briefly outlined the different mechanism of action of many cationic antiviral peptides from different sources [84]. Here, we have summarized the mechanisms of action of human cathelicidin and defensin against a selection of enveloped and non-enveloped viruses that spread by various routes in humans; several of these viruses also have the potential for, or have already caused, a significant outbreak.

#### 3.1.3. Human Rhinoviruses (HRV)

Viral respiratory tract infections are mostly caused by human rhinoviruses (HRV) and have higher rates of mortality in patients who are immunocompromised or have existing pulmonary conditions. HRVs are non-enveloped small viruses. Cathelicidin has been proven to have antiviral potency against many viral pathogens. Researchers have demonstrated the antiviral activity of human cathelicidin LL-37 [97] and other mammalian cathelicidins (ovine cathelicidin, SMAP-29, and porcine cathelicidins, Protegrin-1) in vitro against HRV. LL-37 has a direct action mechanism against the virion and acts on both the virus and the host cell. However, the effect was higher when the virus was exposed to peptide before exposure to infection, instead of treating the human lung epithelial cells with peptide after infection. LL-37 does not induce apoptosis or necrosis in HRV-infected cells, but rather reduces metabolic activity compared to uninfected cells at the same peptide dosage [93]. Defensins might not have direct antiviral activity against HRV. However, few viruses could enable the expression or secretion of defensin subtypes after infecting the host cells, including HRV. This phenomenon has been mostly observed with *β*-defensins, which are inducibly expressed, whereas *α*-defensins are expressed constitutively [96]. New therapies based on antimicrobial peptides are of significant interest considering the current restrictive therapeutic alternatives.

#### 3.1.4. Influenza Virus (IAV)

Anti-influenza activity has been observed with both *α*-defensins and *β*-defensins. However, *α*-defensins showed more potency against the direct inhibition of enveloped Influenza A viruses (IAV). These two defensin subunits might have different mechanisms of action against viruses due to their unique features such as dimerization and expression pattern. The role of *β*-defensins derived from different animals against IAV was reviewed by Kalenik et al. [98]. LL-37 has a distinct mechanism compared to defensin. The incubation of LL-37 with IAV revealed increased potency through direct virus interaction, independent of viral aggregation or the prevention of virus binding to or uptake by cells. LL-37 also interacts differently with surfactant protein-D (SP-D) compared to defensins [99]. SP-D is known as the most potent collectin, inhibiting IAV.

#### 3.1.5. Coronavirus, SARS-CoV-2

Recent studies have confirmed the role of LL-37 in preventing SARS-CoV-2 infections. LL-37 can bind to the S1 domain of the virus and mask the ACE2 receptor, thus limiting infection in the host. Apart from direct antiviral activity, LL-37 might play a therapeutic role in reversing the neutrophil extracellular trap (NET) formation and micro thrombosis, associated with the high mortality rate in COVID-19 [100]. Enteric defensin, HD5, interacts with the ACE2 receptor and blocks the SARS-CoV-2 binding site, and prevents infection [101]. Apart from that neutrophil, *α*-defensin, HNP1, can also inhibit the infection caused by enveloped SARS-CoV-2 virus [102]. Another mechanism is when peptide binds to viral glycoprotein and prevents cell fusion by masking ACE2 receptors, as shown in Figure 9.

#### 3.1.6. Human Immunodeficiency Virus (HIV)

Additionally, LL-37 has the ability to combat HIV-1 and stop the replication of viruses [94]. Vitamin D plays a crucial role in innate immunity. It was found that human macrophages lacking in vitamin D express less cathelicidin in response to stimulation, whereas the inverse was true when endogenous vitamin D was present. According to Campbell et al., incubating macrophages with vitamin D causes autophagy and prevents HIV-1 replication in vitro [104]. LL-37 derived, FK-13, and bovine cathelicidin BMAP-27 derived BMAP-18 can also inhibit HIV-1 activity [105]. Human *α*-defensins, HNP1–4 and human *β*-defensin-2 and human *β*-defensin-3 inhibit HIV infection [61], but the *θ*-defensins homolog in monkey, retrocyclin, showed a significant inhibition of HIV-1 in human CD4+ cells [106].

#### 3.1.7. Ebola Virus

Ebola Virus (EBOV) is a highly pathogenic disease, and this RNA virus has a very high fatality rate across the world. Currently there is no approved drug or vaccine against this disease, and the current treatment options include small-molecule therapies and immunotherapies. Small molecules such as nucleoside analogs, an important class of antiviral agents, are now used to treat EBOV along with human immunodeficiency virus (HIV) infection, hepatitis B virus (HBV), hepatitis C virus (HCV), herpes simplex virus (HSV), and other viral infections [107]. BXC4430 and GS-5734 are among the nucleoside analogs showing in vivo efficacy against EBOV [108,109]. Another mode of treatment is monoclonal antibodies (mAB). These antibodies target the viral glycoprotein, which binds to the NPC1 host receptor. Before attaching to the receptor, the glycoprotein (GP) is cleaved by host cathepsin B and L, causing the subsequent endosomal fusion of the virus and host cell membrane. mAB targets the GP processing and Neimann-Pick C1 (NPC1) receptor pathways, serving as an effective method to inhibit this pathogenic virus. LL-37 and two engineered LL-37 peptides have shown the potential to inhibit EBOV infection. According to Yu et al., these AVPs primarily affect viral infection in their early phases and have no impact on viral replication. Additionally, these powerful inhibitors work to prevent later viral infection phases by inhibiting host cat-B-mediated cleavage of glycoprotein [110]. To our knowledge, defensin has not been found to demonstrate antiviral properties against the Ebola virus.

#### 3.1.8. Zika Virus

There is currently no vaccine or treatment available to pregnant women to combat the Zika virus in the developing fetus. Research shows that human cathelicidin-derived LL-37 fragments have shown potency against this virus. Two fragments, GF-17 corresponding to 17–32 residues of LL-37, the main antimicrobial region, and BMAP-18, showed strong in vitro antiviral potency against this virus. These peptide fragments have also been tested in VERO cells and primary human fetal astrocytes, and showed viral inhibition. Additionally, when the type-I interferon signaling pathway was inhibited it resulted in higher Zika virus infection, which was measured by viral RNA production. From He et al.’s work, it was deduced that GF-17 prevents Zika infection by damaging the viral membrane and causing direct inactivation of the virus, as well as through type-I IFN signaling. BMAP-18 only inhibits the virus through type-I IFN signaling [111]. Zika virus is similar to the dengue virus. According to the findings by Lee et al., the amphibian Yodha peptide showed virucidal activity against all ZIKV and DENV variants. Because of the presence of a hydrophobic N-terminus, this peptide can lyse the outer membrane of the ZIKV virus and disintegrate the viral lipid bilayer [112]. Defensin does not appear to have an antiviral role against these viruses, according to current research.

In addition to these viruses, these human antimicrobial peptides can also have antiviral action on other infection types with known or unknown mechanisms of action yet to be discovered. These indicate that LL-37 and human defensin can both act as antiviral coatings with broad-spectrum action, lesser toxicity, and a lower risk of developing antimicrobial resistance.

## 4. Antiviral Peptides as a Surface Coating

Respiratory viruses such as influenza, SARS-CoV, and MERS-CoV are capable of surviving on surfaces for long periods of time—even months [47]. All of these elements—temperature, relative humidity, surrounding media, and surface type—affect the viability of these viruses. Decontamination of the surfaces is therefore required to stop this indirect viral transmission channel. Hand hygiene, personal protective equipment (PPE), and enhanced surface washing and disinfection in healthcare are some measures intended to avoid the infection of mucosal surfaces and the respiratory system. The design of new surface coating strategies applicable to indoor and outdoor surfaces such as glass doors, door knobs, and handles, automobiles, and healthcare surfaces can prevent viral infection and transmission.

These coatings need to be non-toxic, stable, and non-degradable for an extended period of time, environmentally friendly, and easily applicable to shared spaces. Currently, the coating materials with the ability to inactivate viruses are mostly comprised of inorganic and metallic materials, such as silver, copper, TiO_2_, zinc, etc. These are also added as composites in synthetic and biopolymers to increase their antiviral properties. These materials can be toxic and less environmentally friendly, and have a limited compositional range, making their application expensive [113]. In Section 2, the most recent antiviral surface methods are described. The majority of these methods have the downside of losing their antibacterial effectiveness over time. Additionally, the migration of these inorganic and organic materials can be toxic during contact and eventually might lead to the development of antimicrobial resistance [114]. The usage of virucidal coatings may also be constrained by a variety of issues, including non-universal processing methods, decreased efficacy against particular viruses over prolonged contact, and challenges relating to process scalability and cost. AVPs are a great alternative due to their therapeutic efficacy against a range of viruses and the possibility of sequence modifications that allow flexibility, biocompatibility, and improved selectivity qualities for the target surface [76].

One class of peptides, anti-lipopolysaccharide peptides (SALPs), showed broad-spectrum activity against enveloped viruses with no significant toxicity in vitro. These peptides block the entry of a variety of enveloped viruses by blocking heparan sulfate on the plasma membrane of the host cells, known as the binding site for the pathogens [115]. The earliest viral therapeutics focused on small-molecule drugs. However, recently this way of thinking has changed as nucleic acids, peptides, and antibodies are being looked at in viral diseases. Targeting viral factors is leading to inhibitors with high specificity, potency, and minimal toxicity.

In recent years, with advanced surface modification techniques, it has been possible to immobilize AVPs, conjugate these peptides with the surface, and provide enhanced antimicrobial functionality. These immobilization techniques can be divided in terms of the physical and chemical interaction of the peptide with the surface. Surface adsorption and entrapment of the peptides fall under physical immobilization. In the chemical immobilization method, the peptides are covalently attached or cross-linked to the base surface (Figure 10).

One of the most common techniques of surface functionalization is the covalent immobilization of the peptides on the surface. The immobilization technique can be both selective and non-selective in nature. In selective covalent immobilization, the peptide sequences are modified so that they only bond to the material that makes up the coating. The latter is focused on the natural bond formation with the surface without changing the peptide sequence. The inherent peptide groups, such as amino, carboxyl, hydroxyl, etc., are used in bond formation. However, for the covalent bond to take place, the surface needs to be functionalized with known NHS, PDITC, aldehyde, and similar functionalities [119].

The physical adsorption of peptides on a surface is another widely utilized method. It is known that these important coating strategies behave differently than they do in the soluble peptide phase. The antimicrobial effect of peptides is different when they are in a soluble phase than when they are attached to a surface [119]. This is because the action of peptides depends on their secondary structure. In surface tethered form other surface interactions take place, such as bacterial or viral adherence to the surface, as well as interactions between microorganisms and proteins. Additionally, the type of terminus through which the peptides are bound to the surface and the presence or absence of a linker region (Figure 11) showed different antimicrobial effects [118,120,121].

When taken as a whole, selecting a surface coating method using AVPs is challenging. A number of factors, including the peptide type, its sequence and structure, the type of surface bond formation, surface topology, and base materials, have a considerable impact on choosing the ideal combination for the intended application. Recently, some of these peptides have been used to chemically immobilize medical device surfaces to give them antimicrobial properties [122,123]. On titanium, glass, contact lens, fabric, and stainless steel surfaces, synthetic antimicrobial peptides with active functional groups derived from naturally occurring peptides, such as melamine, E14LKK, synthetic analogs of BMAP-27 and BMAP-28, etc., as well as natural AMPs, such as LL-37, burofin, HDPs, HDs, etc., were immobilized [123,124].

The current and widely used surface coating methods, as well as cutting-edge peptide coating techniques, will be discussed in the parts that follow. The majority of these coating techniques were developed for bacteria, but since natural, synthetic, and recombinant peptides have been known to have potent antiviral properties they will likely work effectively against viruses. In light of the current outbreak, these peptide-based coatings will help to prevent the spread of viral infections and offer more protection.

### 4.1. Covalent Immobilization on Modified Surfaces

Covalent immobilization techniques keep the peptides attached to the surface through covalent bond formation between the peptide and the surface groups. Among selective and non-selective immobilization techniques, non-natural bond forms between the peptide and the surface, achieved by modifying the peptide sequence and providing reactive functionalities. Non-selective methods contain the natural bond formation of peptides without chemically modifying them for attachment. The degree of surface coverage and peptide orientation at the surface has a significant impact on this method’s effectiveness. Physical, chemical, and biological characteristics and specificity of the peptides and the mode of action should be thoroughly investigated before a strategy is developed [38]. Peptides are expensive, and the peptide amount also needs to be optimized to obtain the maximum antimicrobial protection.

BMAP-27, LL-37, and Melittin were used as antimicrobial peptides, and NHS surface treatment and PDITC surface treatment were tested by Rapsch et al. [125]. Surprisingly, the results conclude that the immobilization technique is more important to impart antimicrobial activity than the peptide amount. The resultant connection between the peptide and the surface was different in these immobilization techniques. After optimizing the surface modification technique of the coupling reaction to covalently attach the peptide, the peptide concentration can be optimized to reduce the cost of these peptide-treated surfaces. However, it could be determined that, in order to utilize the antimicrobial capabilities of the covalent surface coating, a threshold concentration of tethered peptide that is dependent on the immobilization approach must be attained.

In a separate study, LL-37 was immobilized on a hydrophobic coating layer via a coupling reaction. The hydrophobic layer was introduced on the surface of magnetic nanoparticles with plasma, well known as a surface modification technique [126]. These surface coatings were tested on mixed cultures and only the viability of *E. coli* was reduced. As LL-37 and BMAP-27, from the human and bovine cathelicidin family, respectively, are known to have broad-spectrum antiviral potency against viral pathogens [127], this coating technique can be confidently included in an antiviral surface design. An antiviral nanocomplex was investigated by Zhang et al., and showed stability at physiological pH and antiviral efficacy against HCV and HIV co-infection. This self-assembled complex was made from electrostatically coupling the cationic HCV NS5A protein to a negatively charged poly(amino acid) based block copolymer [128]

Dhvar5, a synthetic peptide derived from the histatin family, was covalently immobilized on a chitosan-coated thin surface. Chitosan is also known as a natural antimicrobial polysaccharide. This combination of peptide and substrate leads to different results in bacterial adhesion based on the exposed termini (either N or C terminal) to the bacterial supernatant. It is still debatable whether a hydrophobic N-terminus or cationic C-terminus should be used to immobilize the peptides on the surface. According to Hilpert et al., exposing the hydrophobic terminus should have higher antibacterial properties due to more interaction with the lipophilic membrane [120]. However, Chen et al. and Costa et al. immobilized the peptides on the N-terminal tail and found more antiadhesive and bactericidal activity this way [129,130]. The authors concluded that this immobilization technique is highly dependent on the mechanism of action of the peptides on the pathogens. Membrane disruptive peptides should have their hydrophobic end exposed, whereas for the peptides whose activity is conformation-dependent the orientation of immobilization does not matter in those cases. Therefore, the immobilization pattern is unique to each peptide and its target microorganisms and should be thoroughly screened. This covalent immobilization technique has also been tested on different modified surfaces such as glass [131], polystyrene [132], magainin-1, and nisin on stainless steel [133] and bactericidal effects, showing different results between surfaces. The peptide coverage and charge type on the coating surface might be responsible for observing different results [131].

In another study, lyotropic liquid crystal hydrogel (cross-linked DA-F127 bulk hydrogels) has been used as an intermediate substrate to covalently bond cationic peptide, and then the hydrogel particles were subsequently immobilized on PDMS substrate [134]. The hydrogel-based dual-functionalized coating might be used to distribute drugs while also destroying microorganisms on contact. The interpenetrating polymeric networks (IPN) were formed due to the amphiphilicity of the hydrogel and can create a more stable coating than only PDMS alone on implantable medical devices. Many different peptides have been covalently immobilized on unique substrates such as glass or polymer films, for example Melimin and its derivative, Mel4, on a glass surface [135] and synthetic peptide E14LKK on poly(ethylene) film [124]. The review article by Costa et al. provides information on the effects of the covalently immobilizing peptide on surfaces using various chemical coupling techniques, spacer length, and peptide concentration and orientation [121].

Depending on the particular peptide, chemistry, and support employed, different antimicrobial coatings have different modes of action. Additionally, the comparison of all the results points to the possibility that free and immobilized peptides work in a variety of ways. The majority of this study has not yet been tested against viruses. However, it is not implausible to think that these methods will also be effective against viral pathogens given the broad spectrum antiviral activity of AMPs.

### 4.2. Electrostatic and Covalent Attachment of AVPs on Surfaces

In this approach, a dual coating of a defensin-based peptide was tested on a hydroxyapatite (HA) surface. HA surfaces are commonly used as a coating on metal implants to promote bone growth and implant function in patients. The peptides were attached to the surface by covalent bonds, and then more peptides were added that were held in place by electrostatic forces. Most AMPs exhibit a half-life stability of minutes to a few hours when they are in a physiological serum-rich environment [136]. However, this method showed in vitro stability for over 12 months and could prevent Gram-positive and Gram-negative bacteria in simulated body fluid conditions. Thus, this novel method also has a strong possibility of maintaining stability when used in vivo. The covalent layer of the peptide worked as the permanent coating, whereas the electrostatically attached layers worked to release peptides around the tissues, maintaining a sterile environment near the surgical site [137]. This type of coating also served to deter bacterial colonization for a considerable amount of time because defensin is known to kill or prevent the attachment of microorganisms. The short-term distribution of peptides near the implant site, the long-term stability of coated peptides, and the maintenance of the surface properties of the HA material without permanently modifying it are all concerns that this technique has addressed in the field of antimicrobial coating research. Other pathogenic species or surfaces have not been tested for this coating’s efficacy. Given that defensin has antiviral properties, this dual-coating approach may be beneficial in limiting viral adhesion to common healthcare surfaces.

### 4.3. Chemical Vapor Deposition

AMPs have been immobilized on different surfaces such as dental implants, contact lenses, and catheters and have worked as antimicrobial surface coatings using mostly adsorption or chemical modification techniques. However, peptide adsorption has challenges, such as having a complex process of surface modification, uneven peptide distribution on the surface, and the non-selective orientation of peptides. Jeong et al. simplified the process with the one-step immobilization of a synthetic AMP (Cys-linked SHAP1) on different solid surfaces, such as glass slides and latex gloves [123]. They used initiated chemical vapor deposition (iCVD), a solvent-free low-temperature polymerization technique. The surface was vinyl functionalized, which helped the conjugation of the thiol group containing Cys residues of the peptides. These peptides must have a Cys residue at the terminus and UV polymerization for thiol-ene click conjugation. Additionally, the peptides were genetically modified instead of using chemical modification. Glass, latex, polyethylene (PE) film, and paper, among other solid supports, can also be easily functionalized with poly(2,4,6,8-tetravinyl-2,4,6,8-tetramethyl-5-cyclotetrasiloxane) (pV4D4) using the solvent-free iCVD method. Though this method of immobilizing peptide showed high antimicrobial activity against pathogenic bacteria, antiviral efficacy was not tested. Another polymer substrate, such as dibromomaleimide, can also be used in the CVD method to immobilize the cecropinmelittin hybrid antimicrobial peptide [46,138]. This technique has also been applied to coat antimicrobial polymers on fabric [139]. It has to be investigated whether natural polymers such as cathelicidin, defensin, synthetic peptides, or recombinant peptides with established antiviral activities can be immobilized on surfaces using iCVD and can exhibit significant antiviral effects.

### 4.4. Physical Adsorption of AVP

Apart from covalent bonding, another way to immobilize proteins or peptides is via the noncovalent adsorption of peptides on the surface. In a molecular dynamics simulation by Soliman et al., results show that nanoscale surface properties can alter the interaction between the attached peptides and the solid substrate [140]. Hydrophobic interactions kept the structural integrity of the peptide and also showed the strongest adsorption to the surface monolayer. However, electrostatic interactions proved to change the secondary structures of the peptides, which might cause less adsorption capacity and antimicrobial effectiveness [140]. The physical adsorption of AVPs onto surfaces has the advantage of being straightforward, requiring no additional structural modification of the peptide, and having adjustable peptide release kinetics. In this case, surface properties such as roughness, porosity, and surface energy may greatly influence the peptide coating on the substrate, especially on the heterogeneous surface where the distribution is less likely to be uniform. Ye et al. found that the interaction between the peptide and substrate is more dependent on surface polarity than surface charge for the self-assembled, designed GL13K peptide. GL13K is a known peptide that, when dissolved in alkaline solutions, could change the secondary structure into a βsheet and self-assemble into nanofibrils and other metal surfaces. They also experimented with different surfaces, such as negatively charged hydroxyapatite (HA). These amphiphiles were highly adsorbed on the negatively charged, polar HA surface and created a very stable and hydrophobic coating [141].

### 4.5. Layer-by-Layer Adsorption

Layer-by-layer (LbL) assembly uses polyelectrolytes to form a uniform nanometer-thick surface from polyelectrolyte solution with alternating deposition of polyanions and polycations, and provides an infinite adsorption binding site (Figure 12). This method is a potent approach for non-covalent modification in charged surfaces. This assembly has been applied to biomaterials and the controlled release of therapeutic drugs. This method can also be applied to adsorb charged peptides on the surface to demonstrate its inherent antimicrobial efficacy.

A chitosan layer was used as a coating on a surface that exerted efficacy against most known *E. coli* and *S. aureus*. Additionally, LBL nanocoating has found its application in eye lenses to reduce anti-adhesion and bactericidal effects. Apart from polysaccharides and inorganic polyelectrolytes, many proteins such as histone f3, hemoglobin, etc., have been used as polycations, and other enzymes and catalase have been applied as polyanions. DNA can also be used as a polyelectrolyte in this assembly. Many researchers have tried to incorporate AMPs with a carrier for the sustained release of the peptides [143]. One example is using an amphiphilic positively charged peptide, G(IIKK)_4_I-NH_2_, on a negatively charged graphene oxide nanosheet following LBL assembly. This composite can demonstrate antibacterial capability through the controlled release of the peptide [144].

LbL nanocoating of antiviral polysaccharides such as chitosan and its role in the prevention of coronavirus infection have been reviewed elsewhere [145]. Additionally, a controlled release profile of Ponericin G1 peptide from LbL-assembled degradable polyelectrolyte thin films was observed. The release occurred over 10 days in vitro and showed intact antibacterial activity against *S. aureus* over the release period [143]. This versatile LbL assembly can thus be modified to solve problems associated with current antimicrobial surfaces’ short-term effectiveness. Similarly, it is not far-fetched to consider the LbL technique as a surface coating effective against viruses. The key is to carefully select an antiviral peptide and a surface appropriate against the target pathogen.

Researchers have developed a screening method for tethered peptides bound to the surface that retains their antimicrobial activity. Free peptides in solution show different characteristics to tethered peptides. Therefore, these screening models containing four important parameters (hydrophobic ratio, charge, polarity, and hydrophobic portion moment along the length of the sequence) can help prepare a library of cationic peptides conjugated to a specific polymer, or support against a particular pathogen and save time and effort trying to optimize the peptide–surface interactions [120].

### 4.6. Polymeric Brush Coating and Peptide Conjugates

The surface grafting of polymer brushes is a very useful technique to modify the natural properties of a surface. This technique has also been used to conjugate peptides to functionalize surfaces against pathogens. Earlier research has shown that polymer brushes with neutral and zwitterionic properties can help to reduce bacterial attachment to surfaces. After peptides are immobilized on the brushes, they can aid in the antifouling properties of the surface, depending on the hydrophobic–hydrophilic ratio and secondary structures. It might be that polymers can largely influence the peptide structure and change its properties. To reduce the impairment of the peptide, the density of the brushes, structure, surface coverage, and hydrophobic nature of the peptide needs to be thoroughly screened. To create the most effective antibacterial and antifouling surface coating, new coatings using tethered peptide technology should carefully evaluate polymer chemistry and AMP properties [119]. Figure 13 summarizes the recent application of polymer brushe technique in different fields.

This method can be further upgraded to increase functionality, for example by adding an arginineglycine-aspartate (RGD) binding site that will allow the adhesion and proliferation of tissue cells, along with being antiadhesive and antimicrobial. This trifunctional polymeric brush coating was created with Poly(ethylene oxide) chains of the PluronicF-127 (PF127) triblock copolymer, chemically attached at the terminal ends to an AMP or a short RGD peptide moiety [147].

These polymeric brushes are typically attached to the surface by polymerization techniques, requiring precise UV exposure time and subsequently immobilizing the peptides by coupling reaction. However, the lower yield of surface functionalization can be overcome if the peptide–polymer conjugation can be carried out in the solution phase instead of two steps in the surface functionalization, with polymer brushes first and then with the peptide [147]. Another way to further increase the antimicrobial properties of this coating is to use a hydrophilic coating with PEG as a background and graft brushes on top of that [148]. PEG can prevent protein adhesion and, thus, virus attachment at the surface. This layering strategy and changing the surface nano-topography were also tested against pathogenic bacteria. The presence of polymer brushes on hydroxyapatite (HA) nanorods increased the stability and loading of the synthetic peptide, HHC-36, according to Li et al. [149]. Numerous binding sites of polymer brushes are a significant advantage and have a far higher capacity for peptide attachment, resulting in increased antimicrobial properties.

### 4.7. Nanoparticle and Peptide Conjugation

FluPep is a well-known inhibitor of influenza virus infection. According to Alghaair et al., modified FluPep can be coupled with gold and silver nanoparticles to increase its antiviral activity and the IC50 value of the conjugated peptide was noticeably lower than that of free peptide [150]. The antimicrobial properties of silver ions may make it possible to create even more effective antimicrobial inhibitors that can combat both bacterial and influenza co-infections. Numerous studies have been conducted on the effectiveness of nanoparticles containing peptide conjugates against yeast, bacteria, and viruses. Lipids, metals such as silver, gold, titanium, ruthenium, etc., polymers, and hydrogels are only a few examples of the materials that can be used to create nanoparticles (NP). According to Makowski et al.’s review on metal NPs (MNPs) and peptide conjugates, these compounds have broad-spectrum microbiological activity due to the synergistic antimicrobial effect of AMPs and MNPs [75]. Additionally, silver or gold nanoparticles conjugated with mercaptoethane sulfonate (MES) inhibited HSV-1 infection in cell culture by preventing viral attachment, entry, and cell-to-cell dissemination. MES conjugation can mimic heparan sulfate (HS), present on the cell membrane and the primary binding site for many viruses before entry into the cell. MNPs can be replaced with biodegradable protein nanospheres (BSA-based) conjugated with MES, which demonstrated antiviral activity against HSV-1 in the same way as MNP conjugates [151]. Gessner et al. described different conjugation methods of peptides on organic and inorganic NPs [152]. A summary of this technique is presented in Figure 14.

### 4.8. Recombinant Polymer-Based Coatings

Recombinant protein polymers (rPP) are a new type of biopolymer based on nature and use recombinant peptides to enhance the polymer functionality as a surface coating. Recombinant DNA technology is used to produce the peptide along with polymer sequences in microorganisms (Figure 7b). Pereira et al. created new materials based on this technique, SELP-59-A, a silk-elastin-like protein, and A200, an elastin-like recombinamer. These materials contained antimicrobial peptide sequences. They also concluded that the antimicrobial efficacy of these novel materials is attributable to the peptide sequence, not to the backbone polymer structure [78]. De Costa et al. developed a new fusion material, BMAP-18-A200,based film [153]. This film was stable without the addition of any cross-linking agent and showed antibacterial and antifungal properties. Lima et al. described the potential application of nanotechnology-based elastin polymers in various fields [154]. These biodegradable and environmentally friendly techniques can provide a superior solution over synthetic peptide synthesis due to their greater control over peptide sequence. The variety of protein polymers created by genetic engineering is currently somewhat constrained. The cost of this technique depends on downstream purification steps and the initial genetic engineering design of these hybrid peptides. This unique technique can also provide antiviral efficacy depending on the specificity of the peptide in the recombinant polymer sequence.

## 5. Challenges with Peptide Coatings

Despite the obvious benefits of employing these peptides, there are still a number of challenges in the clinical development of peptide-based anti-infection medicines and coatings. The drawbacks include the enzymatic breakdown of peptides in body fluids, the potential toxicity to host tissue cells or surrounding microorganisms, low water-solubility due to the presence of hydrophobic residue, and high synthesis and handling costs. Additionally, the peptides might lose activity when exposed to environmental factors such as high temperature or UV light exposure while immobilizing peptides on the surface. These peptides mostly have smaller amino acid sequences, and synthetic and recombinantly designed peptides contain even smaller sequences. Then, in vitro and in vivo stability validation in a physiologically simulated environment is necessary before designing antiviral peptide-functionalized coatings. In the presence of human blood serum immobilized peptides seem to aggregate, whereas in water peptide structure is not altered [155]. The increased ionic strength of serum thus affects the stability of the peptide [156]. However, higher ionic strength increased the antibacterial properties of the LL-37 peptide [157]. With peptides such as HDPs, a fine balance in concentration is required and above that range the peptides might be toxic to the nearby cells. In therapeutic applications, coating nanoparticles with peptides can improve the distribution, half-life, and lower toxicity of these peptides [75].

Peptide-conjugated surface coatings are proven to be more stable than release-based coatings and free-soluble peptides. Due to the presence of a cleavage site with high arginine and lysine content, these cationic peptides have a shorter half-life of a few hours. This challenge can be mitigated by engineering the arginine residues with *α*-amino-3-guanidino-propionic acid (Agp) [158]. It is promising that RRP9W4N synthetic peptide, when conjugated to an elastin-like polypeptide (ELP) surface, could maintain stability for up to 24 h when incubated in human serum media [155]. Despite these somewhat stable surfaces, the stability of the coating techniques outlined in Section 4 needs to be further studied in order to be improved for practical use. Numerous researchers are attempting to immobilize different antimicrobial peptides on surfaces, including glass, latex, polyethylene, paper, and other materials. The current situation and potential of the usage of all antiviral agents in PPE, face masks, and public areas were summarized by Rakowska et al. [5].

Immobilizing peptides on a surface is not straightforward, as different technical applications have distinct demands for surface materials. Different peptide properties such as specificity against the target virus, orientation during immobilization, selection of spacer or linker, optimal exposure, and not disturbing the secondary structure should be carefully considered. The surface properties, such as porosity, surface charge, energy, etc., can also greatly influence the adhesion and stability of the coating. The functional groups of the peptide play a significant role in selection. As a result, selecting the appropriate immobilization method necessitates taking into account both the material and the peptide being immobilized. It is obvious that both material science and biotechnology will contribute significantly to the creation of novel and realistic coating strategies to contain viral outbreaks. Alongside bacteria, broad-spectrum antiviral approaches should also be researched and investigated thoroughly to help us prepare for and overcome potential viral pandemic challenges in the future.

## 6. Conclusions

Viral outbreaks entail widespread illness, death, and economic disruption. The focus is usually placed on developing antiviral drugs to curb the impact. However, it has come to the knowledge of the scientific community that infected surfaces contribute greatly to the spread of the virus, and necessary measures need to be taken. The application of disinfectants for surfaces in places such as hospitals and public settings is not a long-lasting option. Conventional surface coatings also have limitations, for example cytotoxicity, susceptibility to microbial fouling, etc., that make them unsuitable for healthcare devices and sensitive surfaces.

Therefore, the introduction of novel materials to develop more effective coatings is necessary. This study aims to present an overview and provide a broad understanding of the current state of knowledge and practices in the area of antiviral coatings to identify areas where further research is needed. To develop a benign, environmentally sustainable and cost-effective coating, a rigorous understanding of their working principles as well as their pros and cons is necessary. As discussed above, current antimicrobial coatings have some limitations that can be overcome by utilizing antimicrobial peptides. This has the potential to achieve tunable surface chemistry that renders enhanced efficacy against microorganisms. However, extensive study is required to make it cost-effective for large-scale production and establish it as a sustainable surface coating material.

## Figures and Tables

**Figure 1 viruses-15-00640-f001:**
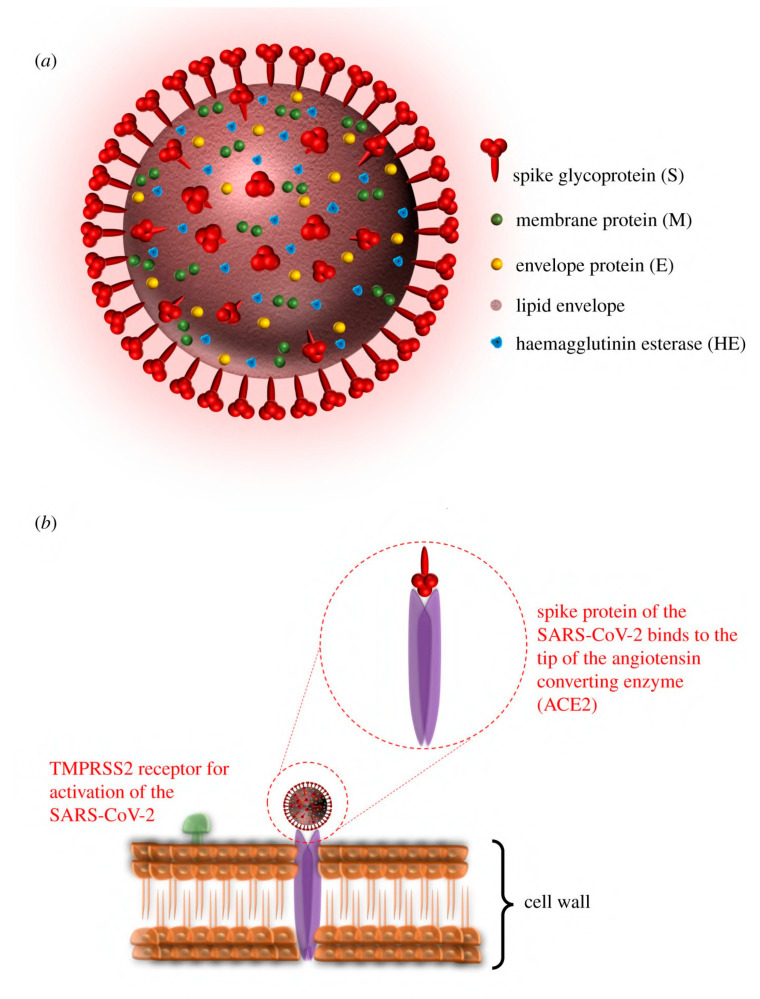
Illustration of (**a**) the structure and proteins of the coronavirus, and (**b**) the mechanism for cell entry via spike proteins (adapted with permission from Aydogdu et al. [10]).

**Figure 2 viruses-15-00640-f002:**
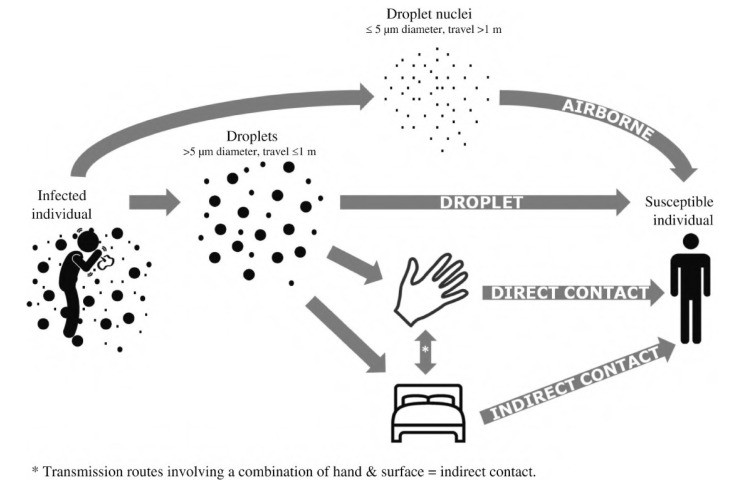
Schematic representation of how respiratory viruses spread through droplets (adapted with permission from Otter et al. [15]).

**Figure 3 viruses-15-00640-f003:**
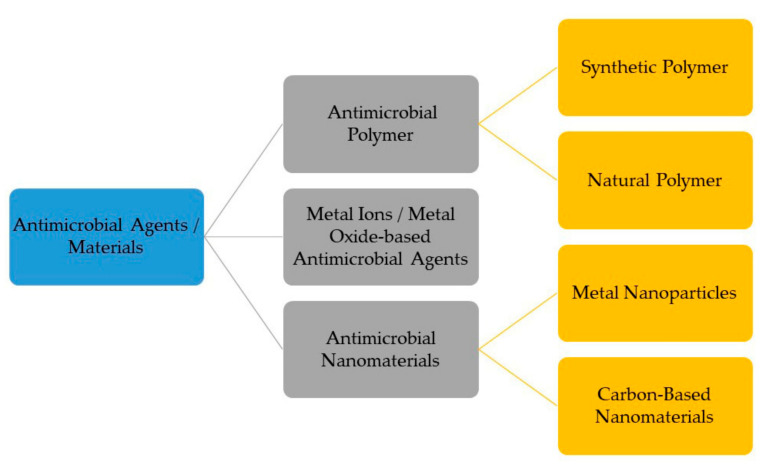
Classification of current antiviral surface coating materials (adapted with permission from Nasri et al. [39]).

**Figure 4 viruses-15-00640-f004:**
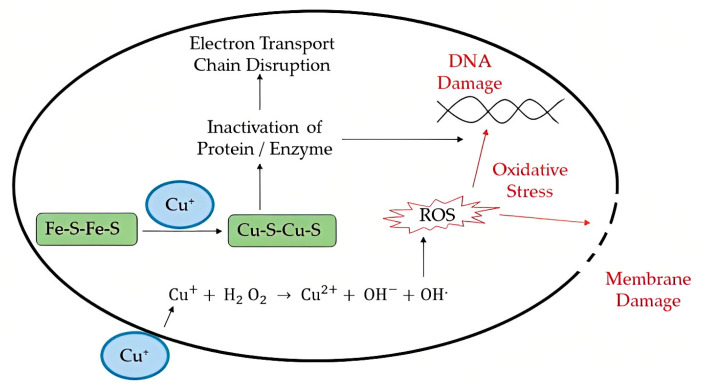
Antiviral mechanism of action of copper ions (reprinted with permission from Nasri et al. [39]).

**Figure 5 viruses-15-00640-f005:**
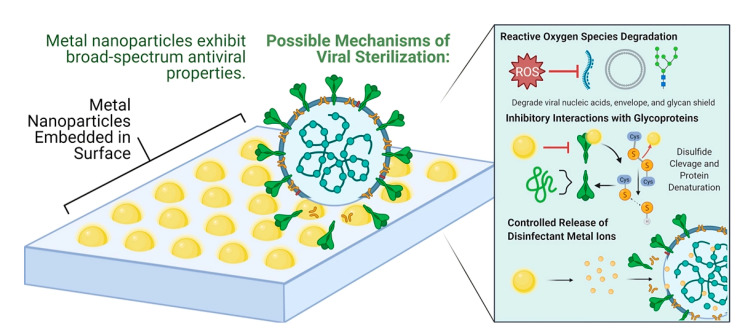
Possible antiviral mechanism of metal nanoparticles (reprinted with permission from Lin et al. [43]).

**Figure 6 viruses-15-00640-f006:**
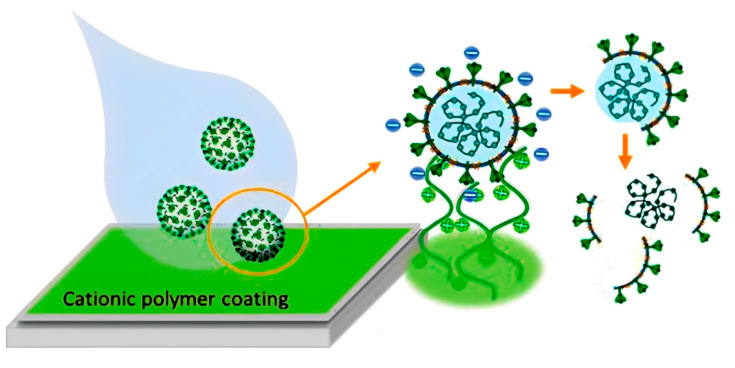
Schematic demonstration of the antiviral mechanism of a cationic polymer (adapted with permission from Mouritz et al. [60]).

**Figure 7 viruses-15-00640-f007:**
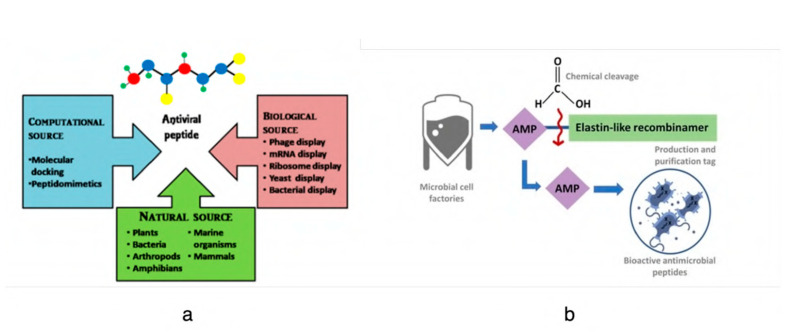
(**a**) Sources of antiviral peptides (reprinted with permission from Agarwal et al., 2021 [77]); (**b**) summary of recombinant antimicrobial peptide production steps. ELR is an elastin-like recombinamer used as a fusion tag to purify the recombinant peptide in the downstream application (reprinted with permission from Pereira et al., 2021 [78]).

**Figure 8 viruses-15-00640-f008:**
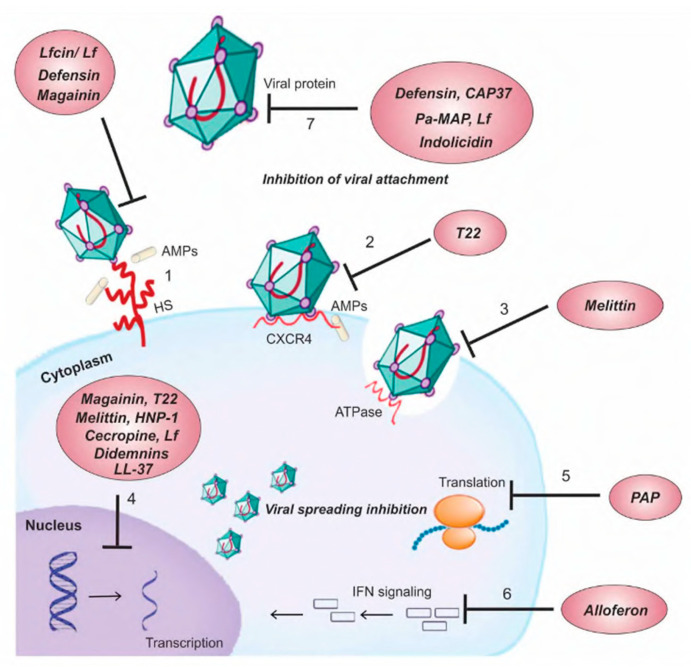
Summary of different mechanisms of action of cationic antiviral peptides from various sources (CAP37, LL-37, defensin, lactoferricin from humans, Alloferon, Melittin, Cecropin from insects, Didemnins from tunicate, Magainin from frogs, Pa-MAP from fish, PAP from plants, synthetic peptide, T22). Targets on the cell surface: (1) Peptide interactions with various glycosaminoglycans that compete with the virus for cellular binding sites. (2) The peptide stops the virus from getting into the cell by attaching itself to the viral receptor. (3) Avoiding cell fusion by preventing ATPase protein activity. Targets inside cells: (4) Prevention of viral gene expression, (5) ribosome inactivation to prevent peptide chain elongation, (6) induction of NK and IFN to activate an immune modulatory pathway. Viral protein targets: (7) Inhibition of adsorption/virus-cell fusion caused by peptide binding to viral proteins (reprinted with permission from Mulder et al., 2013 [84]).

**Figure 9 viruses-15-00640-f009:**
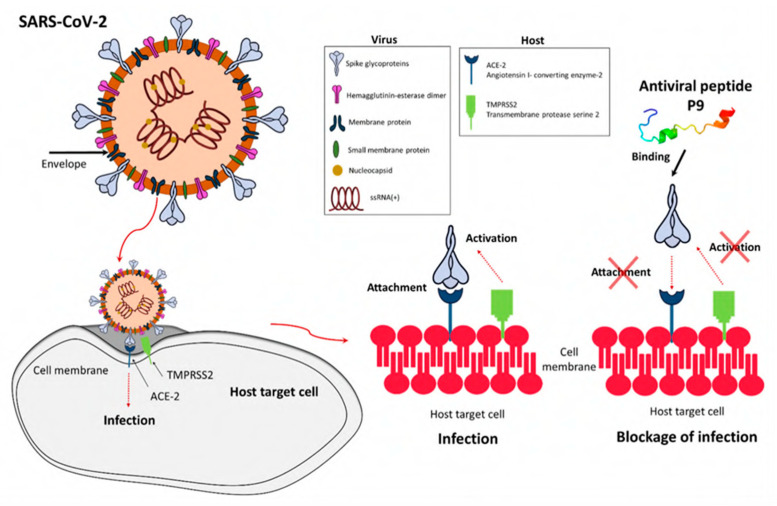
Antiviral peptide P9, derived from mouse *β*-defensin, prevents SARS-CoV-2 infection by attaching to viral glycoprotein and masking ACE2 receptor on host cells (reprinted with permission from Tonk et al., 2021 [103]).

**Figure 10 viruses-15-00640-f010:**
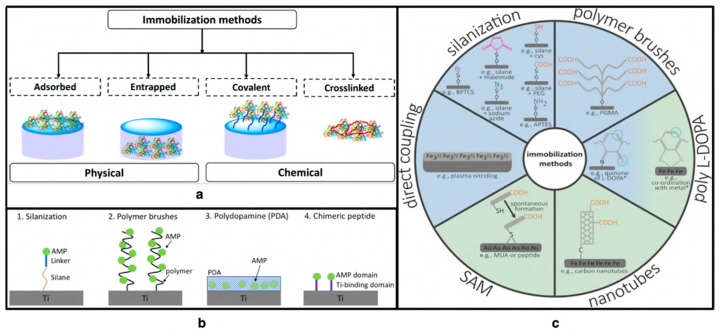
(**a**) Different peptide immobilization techniques on a solid surface (reprinted with permission from Imam et al. [116]). (**b**) Covalently immobilizing peptides on Titanium (Ti) surface following different techniques (reprinted with permission from Andrea et al. [117]). (**c**) Some other covalent immobilization techniques for biomedical application (reprinted with permission from Stillger et al. (* possible binding mechanisms are discussed in “Polymerization of l-DOPA” section in ref. [118]).

**Figure 11 viruses-15-00640-f011:**
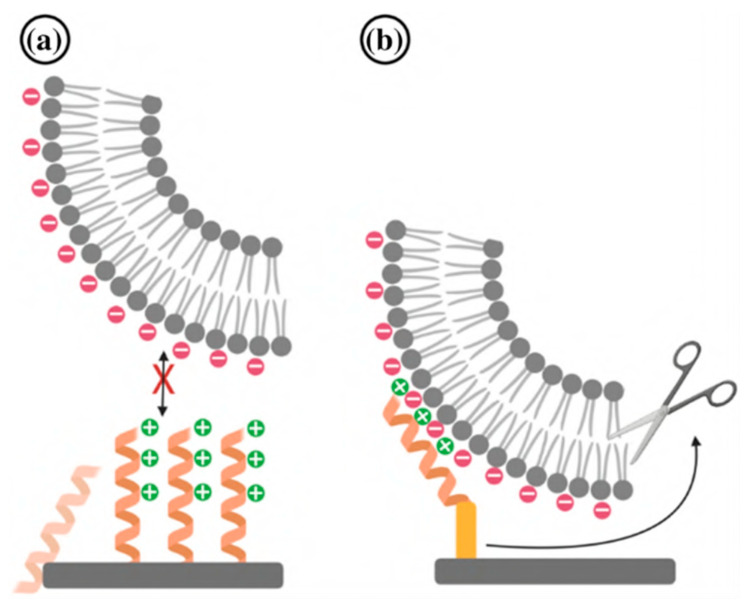
(**a**) Rigid or tight immobilization of cationic peptide prevents interaction with pathogen membrane. (**b**) The Linker region (yellow) provides flexibility to the peptides to form a secondary structure and target the anionic membrane when in contact (reprinted with permission from Stillger et al. [118]).

**Figure 12 viruses-15-00640-f012:**
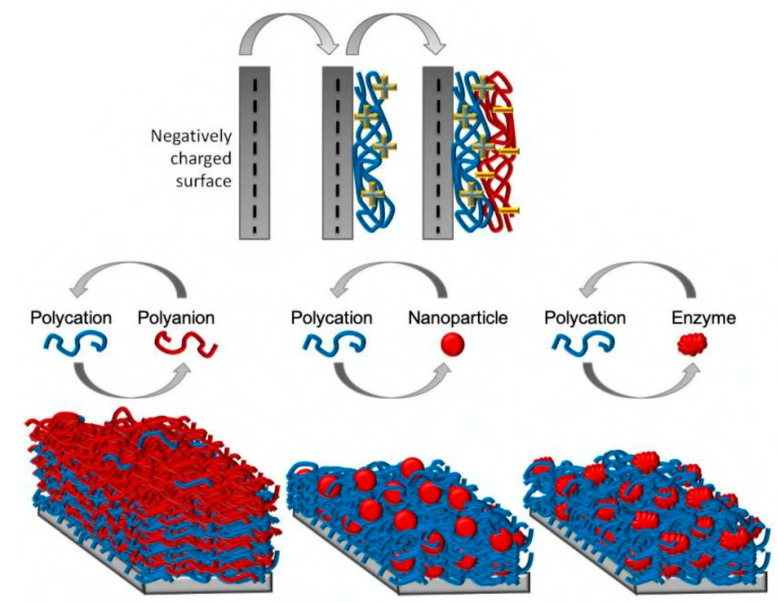
Layer-by-layer coating technique. Positive and negative charge molecules are adsorbed one after the other on the surface, and this cycle continues n times to obtain the desired film thickness (reprinted with permission from Escobar et al., 2021 [142]).

**Figure 13 viruses-15-00640-f013:**
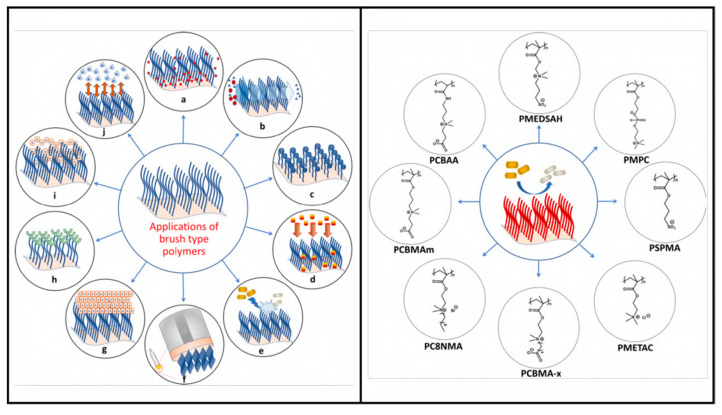
Applications of brush-type polymer coatings (a; drug delivery systems, b; membrane applications, c; smart membrane and surface applications, d; biosorption, e; antibacterial surfaces, f; biosensor applications, g; cell adhesive surfaces, h; biocompatible coatings, i; tissue engineering applications, and j; antifouling surfaces) (**Left**). Brush-type polymer structures are commonly used to prevent microbes from adhering to surfaces (**Right**) (reprinted with permission from Açarı et al., 2022 [146]).

**Figure 14 viruses-15-00640-f014:**
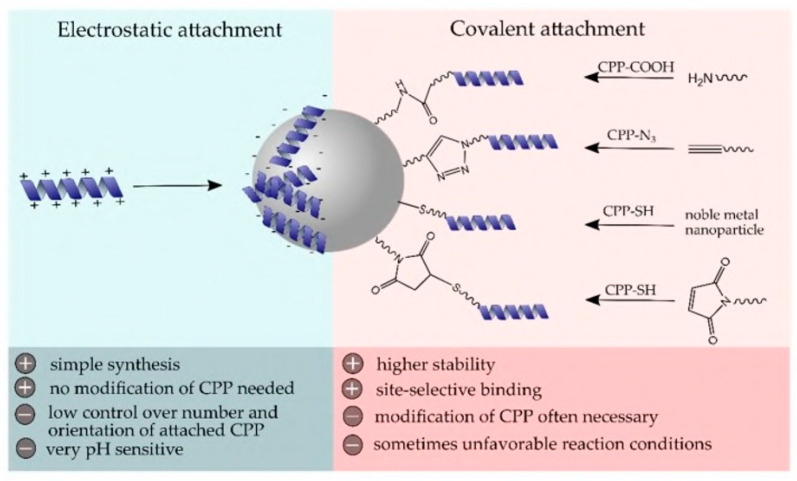
Conjugation of peptides on nanoparticle (NP) by electrostatic immobilization and covalent immobilization (adapted with permission from Gessner et al. [152]).

## Data Availability

In this review paper, no new data was generated.

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
