# Peer review of "Antiviral Peptides in Antimicrobial Surface Coatings—From Current Techniques to Potential Applications"

_viruses, 2023, doi:10.3390/v15030640_

Round 1

Reviewer 1 Report

The authors made a very fruitful effort for this comprehensive review; however, some parts need more attention as follows-

1. The introduction needs to be updated, say the authors mentioned viruses and their characteristics very briefly in the first two paragraphs. Still, only one is dedicated to the peptides and the modifications need at least other paragraphs with brief coverage of peptides, modification, coating, and conjugation. 

2. The whole manuscript is missing the proper representative diagrams. Please include the relevant diagrams or representative figures in the sections and sub-sections. 

3. In the 4.7 section missing a proper discussion about the nanoparticles and peptide conjugation, not only silver and gold but there are also numerous kinds of nanoparticles which are coming out as an alternative like Titanium, Ruthenium, authors must highlight broad area particles also.

4. The article is missing the metal and organic complex peptide conjugation parts in the subsection of the number 4, they must include this section in the review.

Author Response

Reviewer 1:

The authors made a very fruitful effort for this comprehensive review; however, some parts need more attention as follows-

Reply: We sincerely thank the reviewer for his/her feedback.

  1. The introduction needs to be updated, say the authors mentioned viruses and their characteristics very briefly in the first two paragraphs. Still, only one is dedicated to the peptides and the modifications need at least other paragraphs with brief coverage of peptides, modification, coating, and conjugation. 

Reply: We have introduced new paragraphs on virus characterization, antiviral peptide, and peptide-based coating conjugation in the introduction based on the reviewer’s comment.

  1. The whole manuscript is missing the proper representative diagrams. Please include the relevant diagrams or representative figures in the sections and sub-sections. 

Reply: The reviewer raised an excellent point here. We have included 14 figures section-wise in the modified version to make the article more readable. 

  1. In the 4.7 section missing a proper discussion about the nanoparticles and peptide conjugation, not only silver and gold but there are also numerous kinds of nanoparticles which are coming out as an alternative like Titanium, Ruthenium, authors must highlight broad area particles also.

Reply: Modified according to the reviewer’s comment in section 4.7

  1. The article is missing the metal and organic complex peptide conjugation parts in the subsection of the number 4, they must include this section in the review.

Reply: The reviewer raised an important point here. Unfortunately, we did not find articles on metal and organic complex peptide conjugation used as a surface coating. Hence, we could not add this in section 4.

Reviewer 2 Report

The review manuscript by Jabeen et al, entitled “Emerging applications for antiviral peptides in antimicrobial surface coating”, provides a general overview on the antiviral capability of antimicrobial peptides (AMPs), as well as recent advances in the use of these peptides for the development of antiviral surfaces.

Although there are several reviews focusing on AMPs, the present manuscript have the potential to fill a literature gap by focusing on the specific thematic of antiviral peptides and associated coatings. Although it addresses the topic in the broad sense, the manuscript should be revised and adapted to present a logical structure (information seems “chaotic” without a connecting flow). Also, the document should be thoroughly revised and improved for clarity and coherency, as it presents several grammar errors, questionable and unnecessary information and severe inconsistencies that need to be addressed.

Main comments:

For a better comprehension and to improve clarity for the readers, it is suggested to include a figure depicting the general classification and structure of the viruses with identification of the structures (or replication process) in which the AMPs act on, as well as the surface charge. Also, a figure depicting the general mechanism of action of AVPs would improve the reading.

Although the subject of the manuscript is on antiviral peptides, the authors tend to expand to the antibacterial relevance of AMPs. I would suggest to try to condense the text on the antibacterial activity. Similarly, the description of the strategies used to obtain antiviral surface coatings, other than antiviral peptides (Section 3), seems too long and deviates from the main topic, which are AVPs. As such, I would suggest to either condense the text on section 3 or adapt the title to meet the review topics; for instance, “Antiviral peptides in antimicrobial surface coating – from current techniques to potential applications”.

In Section 2.4, the authors state that AVPs can be obtained from synthetic or natural sources. However, another route is through the use of recombinant DNA technology. The use of biotechnological tools presents the advantage of a potentially more eco-friendly approach than traditional chemical synthesis, overcoming the limitations of the latter such as cost, scalability and polydispersity, and provide the means to have absolute control over the sequence. This should be mentioned.

Following the previous comment, although different immobilization/adsorption methods are described in Section 4, I miss description on recombinant antimicrobial peptides for the development of antimicrobial surfaces (recombinant proteins engineered at the DNA level to incorporate antimicrobial peptide sequences). This represents an alternative approach to obtaining functional polymers that are fully composed of amino acids and therefore, intrinsically biodegradable and environmental friendly; especially considering that these meet at some extent the criteria for the selection of ideal materials for antiviral surface coatings, and, as mentioned above, overcome the limitation associated with chemical synthesis (e.g. cost-efficiency, monodispersity, complete control over sequence and size, more eco-friendly approach, etc). Some good works can be found, for example, in:

https://doi.org/10.3390/app11125352

https://doi.org/10.1021/acsbiomaterials.0c01262

https://doi.org/10.3389/fnano.2022.874790

https://doi.org/10.3390/ph14100956

https://doi.org/10.1039/D0BM00155D

https://doi.org/10.1016/j.actbio.2019.09.004

https://onlinelibrary.wiley.com/doi/full/10.1002/mabi.201800262

https://iopscience.iop.org/article/10.1088/1748-605X/aa7104

https://doi.org/10.1021/bm5016706

https://www.sciencedirect.com/science/article/pii/S0142961211002031

Finally, the manuscript should be thoroughly and carefully revised to correct typo errors (e.g. dots missing at the end of sentences, extra space between the last word and references, italicize species identification, use subscript numbers in chemical formulations, references missing/misspelled, etc).

Specific comments:

In Abstract

1. in vivo and ex vivo should be italicized (in vivo and ex vivo);

2. line 16, correct “phighlightsome”.

In section 1. Introduction

3. lines 56-59, these sentences should be revised for improved clarity. The authors refer that «silver nanoparticles are a different type of antibacterial agent». This sentence appears to be out of context. First, what the authors mean by “different type” and compared with what? The authors should provide a proper context. Secondly, the authors mention “antibacterial agent” but the focus is on antiviral agents – what’s the relation? In the next sentence (line 58), the authors mention that «silver nanoparticles are harmful to other life forms, making this process time-consuming and complex». There is no relation of causality between being harmful and the process being time-consuming and complex;

4. lines 65-66, again, the authors refer Cu and Ag. However, as mentioned above, the use of Ag as antiviral agent is not clear and thus, these examples seem displaced from the topic context.

In section 2. Peptides and Antiviral Activity

5. Although the subject of the manuscript is on antiviral peptides, and the section title is Antiviral Activity, the authors tend to expand to the antibacterial relevance of AMPs. I would suggest to try to condense the text on the antibacterial activity (especially in subsection 2.1).

6. line 74, correct “moleculess”;

7. line 75, antimicrobial peptides are also present in bacteria (commonly termed as bacteriocins). In fact, this is clearly stated in the reference used by the authors (ref [26]). The assumption that AMPs occur only in multicellular organisms is not correct;

8. lines 90-91, revise as in the present form contains redundant information with lines 74-75. For instance, to «In the skin, AMPs play a crucial role in the innate immune system’s defense against pathogens [30]»;

9. lines 95-97, the authors advocate that AMPs are advantageous over immunoglobulin-based antibodies due to the lower size. The authors should elaborate more on this statement. Also, it is claimed that host cells take less time and energy to make AMPs than for exogenous vaccines to produce antibodies. The purpose of this statement is not clear or logical – what is the relation between taking less time and energy to produce AMPs with the exogenous vaccines? AMPs are part of the innate immune (some expressed constitutively) and vaccines are given to trigger the immune response. Finally, these two sentences should be revised as there is no logical relationship between the first and the second.

10. lines 132-139, references are missing to support the statements;

11. line 140, gram positive and gram negative should be hyphenated;

12. line 142, correct “Cathelicidn”;

13. line 144, provide some examples of the inactivation of biological activity (e.g. at the level of which structures)

14. line 147, S. aureus should be italicized and first reference should be the full name;

15. line 148, correct lacteferrin to lactoferrin;

16. line 152, the authors indicate that the mechanism of action against viruses depend on the type of pathogen. In my opinion, the mechanism of action may depend on the type of virus and not pathogen - in a broad sense, pathogen is an organism causing disease to the host; thus, the pathogens are the viruses. I would suggest to remove this sentence and adapt the next phrase accordingly.

17. subsection 2.2, revise the text to include the greek letter for defensin identification (e.g. lines

164, 165, 169, 184, etc);

18. line 180 is missing;

19. the reason for selecting the six virus-associated diseases in Section 2.3 should be stated or made more clear.

20. line 191, I would suggest to change “potential” to “potency”;

21. line 237, correct dsease;

In section 3. Current Antiviral Surface Coating Techniques

22. line 347, correct Ag4O4;

23. line 353, correct SiO2/Ag;

24. line 356, correct However, Exploiting;

25. lines 357-358, the authors mention “Metal nanoparticles are toxic and are not meant for internal digestion (…)”. Since the subject is on antiviral coatings, what is the relevance of toxicity after digestion?

26. subsection 3.2 “Polymer-Based”, this section be carefully and thoroughly revised as it presents several inconsistencies. For instance, the authors give a description on cationic polymeric coatings, mentioning that the electropositively charged groups in the polymer chain kill microorganisms through a contact-dependent mechanism or impede microbial adhesion, without releasing chemical agents. Figure 2 does not relate with this information at different levels: i) the charges presented in the figure are negative and not positive; ii) as mentioned, polymer-based coatings do not involve a chemical release (antiviral drug).

A arge part of the text refers to the cationic nature of polymers and how charge relates with the antimicrobial activity. Although the antimicrobial action against bacteria is intuitively inferred due to the negatively charged bacterial cell membranes, nothing is said on the charge of viruses and how electropositive polymers exert an action.

Also, the authors give a lot of attention in this section to the antibacterial/antifungal properties and little is said about the state of the art on viruses;

27. lines 366-368, the different categories of antimicrobial polymers should be mentioned;

28. lines 377-379, in my view, antimicrobial peptides have evolved and not survived evolution. This sentence should be rephrased;

29. line 382, indicate the charge of human/mammalian cells;

30. lines 384, remove unnecessary information on hydrogels;

31. line 392, correct cross link to cross-link, throughout the text;

32. lines 394-395, Staphylococcus aureus and Candida albicans should be italicized;

33. lines 426-427, the reason why surfactants are mostly used against enveloped viruses should be elucidated. As mentioned above, including a figure depicting the viral structure will aid in this;

34. subsection 3.4, the criteria for the selection of ideal materials for antiviral surface coating should be introduced earlier. The antimicrobial mechanism of the natural coatings should be elucidated;

35. lines 460-462, I suggest removing the sentence as it is redundant with the last sentence in Introduction;

In section 4. AVPs as surface coating

36. The authors adequately introduce the topic and justify the reason for reporting mostly antibacterial effects. However, the authors should avoid the use of AMPs and AVPs indiscriminately and try to use uniform language throughout the text. Moreover, this section would benefit from a figure depicting the different strategies for AVP immobilization/adsorption;

37. lines 464-520, it is advisable to divide the text into 2 or 3 paragraphs to improve clarity;

38. lines 464-475, in my view, the introduction on viruses is more adequate for the early sections to avoid redundancy;

39. lines the authors refer to SARS-CoV however, it is not in the list of pathogenic viruses from section 2.3;

40. line 478, correct biopolymenrs;

41. line 482, correct in-vitro to in vitro;

42. lines 499-504, revise the text to improve language and clarity;

43. line 503, correct In-vivo to in vivo,;

44. line 504, it is not clear what is meant by “also affect our understanding of the hydrophobic residue”;

45. lines 505-506, include some references;

46. lines 537-542, the text seems unformatted;

47. section 4.3, line 564, indicate reference after Rapsch et al.;

48. line 590, E. coli should be italicized;

49. lines 589-592, I would suggest dividing the sentence in two;

50. line 602, correct and include reference to hilpert et al.;

51. line 604, reference to Chen et al. is missing;

52. line 638, include reference to Soliman et al. immediately after or at the end of the sentence;

53. line 659, E. coli and S. aureus should be italicized;

54. line 711, missing reference to Li et al.;

55. lines 741-742, correct in-vitro and in-vivo to in vitro and in vivo;

56. line 745, how was reached the conclusion that the ionic strength affects the stability of the peptide due to enzymatic degradation? The ionic strength may interfere with peptide structure stability but is doubtful that will result in enzymatic degradation.

Author Response

Reviewer 2:

The review manuscript by Jabeen et al, entitled “Emerging applications for antiviral peptides in antimicrobial surface coating”, provides a general overview on the antiviral capability of antimicrobial peptides (AMPs), as well as recent advances in the use of these peptides for the development of antiviral surfaces.

Although there are several reviews focusing on AMPs, the present manuscript have the potential to fill a literature gap by focusing on the specific thematic of antiviral peptides and associated coatings. Although it addresses the topic in the broad sense, the manuscript should be revised and adapted to present a logical structure (information seems “chaotic” without a connecting flow). Also, the document should be thoroughly revised and improved for clarity and coherency, as it presents several grammar errors, questionable and unnecessary information and severe inconsistencies that need to be addressed.

Reply: We sincerely thank the reviewer for his/her feedback. We have revised the flow and content of the manuscript based on the comments here.

Main comments:

For a better comprehension and to improve clarity for the readers, it is suggested to include a figure depicting the general classification and structure of the viruses with identification of the structures (or replication process) in which the AMPs act on, as well as the surface charge. Also, a figure depicting the general mechanism of action of AVPs would improve the reading.

Reply: We have included 14 figures section-wise in the modified version.

Although the subject of the manuscript is on antiviral peptides, the authors tend to expand to the antibacterial relevance of AMPs. I would suggest to try to condense the text on the antibacterial activity. Similarly, the description of the strategies used to obtain antiviral surface coatings, other than antiviral peptides (Section 3), seems too long and deviates from the main topic, which are AVPs. As such, I would suggest to either condense the text on section 3 or adapt the title to meet the review topics; for instance, “Antiviral peptides in antimicrobial surface coating – from current techniques to potential applications”.

Reply: We have condensed bacterial part in section 2, 3 and 4.

In Section 2.4, the authors state that AVPs can be obtained from synthetic or natural sources. However, another route is through the use of recombinant DNA technology. The use of biotechnological tools presents the advantage of a potentially more eco-friendly approach than traditional chemical synthesis, overcoming the limitations of the latter such as cost, scalability and polydispersity, and provide the means to have absolute control over the sequence. This should be mentioned.

Reply: The authors have highlighted an exciting coating technique. We have briefly touched on this AVP source in section 3.

Following the previous comment, although different immobilization/adsorption methods are described in Section 4, I miss description on recombinant antimicrobial peptides for the development of antimicrobial surfaces (recombinant proteins engineered at the DNA level to incorporate antimicrobial peptide sequences). This represents an alternative approach to obtaining functional polymers that are fully composed of amino acids and therefore, intrinsically biodegradable and environmental friendly; especially considering that these meet at some extent the criteria for the selection of ideal materials for antiviral surface coatings, and, as mentioned above, overcome the limitation associated with chemical synthesis (e.g. cost-efficiency, monodispersity, complete control over sequence and size, more eco-friendly approach, etc). Some good works can be found, for example, in:

https://doi.org/10.3390/app11125352

https://doi.org/10.1021/acsbiomaterials.0c01262

https://doi.org/10.3389/fnano.2022.874790

https://doi.org/10.3390/ph14100956

https://doi.org/10.1039/D0BM00155D

https://doi.org/10.1016/j.actbio.2019.09.004

https://onlinelibrary.wiley.com/doi/full/10.1002/mabi.201800262

https://iopscience.iop.org/article/10.1088/1748-605X/aa7104

https://doi.org/10.1021/bm5016706

https://www.sciencedirect.com/science/article/pii/S0142961211002031

Reply: The reviewer has shared many helpful articles on this topic. Based on the feedback provided by the reviewer, we have included a new subsection 4.8.

Finally, the manuscript should be thoroughly and carefully revised to correct typo errors (e.g. dots missing at the end of sentences, extra space between the last word and references, italicize species identification, use subscript numbers in chemical formulations, references missing/misspelled, etc).

Reply: We have carefully revised the manuscript for these inaccuracies.

Specific comments:

In Abstract

  1. in vivo and ex vivo should be italicized (in vivo and ex vivo);

Reply: Addressed as per the reviewer’s comment in the abstract of the revised manuscript.

  1. line 16, correct “phighlightsome”.

Reply: Addressed as per the reviewer’s comment in the abstract of the revised manuscript.

In section 1. Introduction

  1. lines 56-59, these sentences should be revised for improved clarity. The authors refer that «silver nanoparticles are a different type of antibacterial agent». This sentence appears to be out of context. First, what the authors mean by “different type” and compared with what? The authors should provide a proper context. Secondly, the authors mention “antibacterial agent” but the focus is on antiviral agents – what’s the relation? In the next sentence (line 58), the authors mention that «silver nanoparticles are harmful to other life forms, making this process time-consuming and complex». There is no relation of causality between being harmful and the process being time-consuming and complex;

Reply: Necessary modifications has done as per the reviewer’s comment in the introduction of the revised manuscript.

  1. lines 65-66, again, the authors refer Cu and Ag. However, as mentioned above, the use of Ag as antiviral agent is not clear and thus, these examples seem displaced from the topic context.

Reply: The use of Ag and Cu has been explained in line with the context in the introduction of the revised manuscript.

In section 2. Peptides and Antiviral Activity

  1. Although the subject of the manuscript is on antiviral peptides, and the section title is Antiviral Activity, the authors tend to expand to the antibacterial relevance of AMPs. I would suggest to try to condense the text on the antibacterial activity (especially in subsection 2.1).

Reply: Addressed as per the reviewer’s comment in section 3 of the revised manuscript.

  1. line 74, correct “moleculess”;

Reply: Addressed as per the reviewer’s comment in section 3 of the revised manuscript.

  1. line 75, antimicrobial peptides are also present in bacteria (commonly termed as bacteriocins). In fact, this is clearly stated in the reference used by the authors (ref [26]). The assumption that AMPs occur only in multicellular organisms is not correct;

Reply: Addressed as per the reviewer’s comment in section 3 of the revised manuscript.

  1. lines 90-91, revise as in the present form contains redundant information with lines 74-75. For instance, to «In the skin, AMPs play a crucial role in the innate immune system’s defense against pathogens [30]»;

  1. lines 95-97, the authors advocate that AMPs are advantageous over immunoglobulin-based antibodies due to the lower size. The authors should elaborate more on this statement. Also, it is claimed that host cells take less time and energy to make AMPs than for exogenous vaccines to produce antibodies. The purpose of this statement is not clear or logical – what is the relation between taking less time and energy to produce AMPs with the exogenous vaccines? AMPs are part of the innate immune (some expressed constitutively) and vaccines are given to trigger the immune response. Finally, these two sentences should be revised as there is no logical relationship between the first and the second.

Reply: Removed lines 95-97 as per the reviewer’s comment in section 3 of the revised manuscript.

  1. lines 132-139, references are missing to support the statements;

Reply: Addressed as per the reviewer’s comment in section 3 of the revised manuscript.

  1. line 140, gram positive and gram negative should be hyphenated;

Reply: Addressed as per reviewer’s comment in the revised manuscript.

  1. line 142, correct “Cathelicidn”;

Reply: Addressed as per the reviewer’s comment in the revised manuscript.

  1. line 144, provide some examples of the inactivation of biological activity (e.g. at the level of which structures)

Reply: Addressed as per the reviewer’s comment in section 3 of the revised manuscript.

  1. line 147, S. aureus should be italicized and first reference should be the full name;

Reply: Removed this line to condense the bacterial part and modified in rest of the manuscript.

  1. line 148, correct lacteferrin to lactoferrin;

Reply: Removed this line to condense the bacterial part as per the reviewer’s suggestion.

  1. line 152, the authors indicate that the mechanism of action against viruses depend on the type of pathogen. In my opinion, the mechanism of action may depend on the type of virus and not pathogen - in a broad sense, pathogen is an organism causing disease to the host; thus, the pathogens are the viruses. I would suggest to remove this sentence and adapt the next phrase accordingly.

Reply: Addressed as per the reviewer’s comment in section 3 of the revised manuscript.

  1. subsection 2.2, revise the text to include the greek letter for defensin identification (e.g. lines 164, 165, 169, 184, etc);

Reply:  Addressed as per the reviewer’s comment in section 3 of the revised manuscript.

  1. line 180 is missing;

Reply: Addressed as per the reviewer’s comment in the revised manuscript.

  1. the reason for selecting the six virus-associated diseases in Section 2.3 should be stated or made more clear.

Reply: We have addressed the reasons for selecting these 6 viruses.

  1. line 191, I would suggest to change “potential” to “potency”;

Reply: Addressed as per the reviewer’s comment in the revised manuscript.

  1. line 237, correct dsease;

Reply: Addressed as per the reviewer’s comment in the revised manuscript.

In section 3. Current Antiviral Surface Coating Techniques

  1. line 347, correct Ag4O4;

Reply: Addressed as per the reviewer’s comment in the revised manuscript.

  1. line 353, correct SiO2/Ag;

Reply: Addressed as per the reviewer’s comment in the revised manuscript.

  1. line 356, correct However, Exploiting;

Reply: Addressed as per the reviewer’s comment in the revised manuscript.

  1. lines 357-358, the authors mention “Metal nanoparticles are toxic and are not meant for internal digestion (…)”. Since the subject is on antiviral coatings, what is the relevance of toxicity after digestion?

Reply: Necessary modifications has done as per the reviewer’s comment in section 3 of the revised manuscript.

  1. subsection 3.2 “Polymer-Based”, this section be carefully and thoroughly revised as it presents several inconsistencies. For instance, the authors give a description on cationic polymeric coatings, mentioning that the electropositively charged groups in the polymer chain kill microorganisms through a contact-dependent mechanism or impede microbial adhesion, without releasing chemical agents. Figure 2 does not relate with this information at different levels: i) the charges presented in the figure are negative and not positive; ii) as mentioned, polymer-based coatings do not involve a chemical release (antiviral drug).

Reply: Necessary modifications has done as per the reviewer’s comment in section 3 of the revised manuscript.

A arge part of the text refers to the cationic nature of polymers and how charge relates with the antimicrobial activity. Although the antimicrobial action against bacteria is intuitively inferred due to the negatively charged bacterial cell membranes, nothing is said on the charge of viruses and how electropositive polymers exert an action.

Reply: Necessary modifications has done as per the reviewer’s comment in section 3 of the revised manuscript.

Also, the authors give a lot of attention in this section to the antibacterial/antifungal properties and little is said about the state of the art on viruses;

Reply: Necessary modifications has done as per the reviewer’s comment in section 3 of the revised manuscript.

  1. lines 366-368, the different categories of antimicrobial polymers should be mentioned;

Reply: Addressed as per the reviewer’s comment in the revised manuscript.

  1. lines 377-379, in my view, antimicrobial peptides have evolved and not survived evolution. This sentence should be rephrased;

Reply: Addressed as per the reviewer’s comment in the revised manuscript.

  1. line 382, indicate the charge of human/mammalian cells;

Reply: Addressed as per the reviewer’s comment in the revised manuscript.

  1. lines 384, remove unnecessary information on hydrogels;

Reply: Addressed as per the reviewer’s comment in the revised manuscript.

  1. line 392, correct cross link to cross-link, throughout the text;

Reply: Addressed as per the reviewer’s comment in the revised manuscript.

  1. lines 394-395, Staphylococcus aureus and Candida albicans should be italicized;

Reply: Addressed as per the reviewer’s comment in the revised manuscript.

  1. lines 426-427, the reason why surfactants are mostly used against enveloped viruses should be elucidated. As mentioned above, including a figure depicting the viral structure will aid in this;

Reply: Necessary modifications has done as per the reviewer’s comment in section 3 of the revised manuscript.

  1. subsection 3.4, the criteria for the selection of ideal materials for antiviral surface coating should be introduced earlier. The antimicrobial mechanism of the natural coatings should be elucidated;

Reply: Necessary modifications has done as per the reviewer’s comment in section 3 of the revised manuscript.

  1. lines 460-462, I suggest removing the sentence as it is redundant with the last sentence in Introduction;

Reply: Addressed as per the reviewer’s comment in the revised manuscript.

In section 4. AVPs as surface coating

36 a. The authors adequately introduce the topic and justify the reason for reporting mostly antibacterial effects. However, the authors should avoid the use of AMPs and AVPs indiscriminately and try to use uniform language throughout the text.

Reply: Addressed as per the reviewer’s comment in the revised manuscript.

36 b. Moreover, this section would benefit from a figure depicting the different strategies for AVP immobilization/adsorption;

Reply: Addressed as per the reviewer’s comment in the revised manuscript.

  1. lines 464-520, it is advisable to divide the text into 2 or 3 paragraphs to improve clarity;

Reply: This paragraph has been condensed and the content were moved to introduction.

  1. lines 464-475, in my view, the introduction on viruses is more adequate for the early sections to avoid redundancy;

Reply: This paragraph has been condensed and the content were moved to introduction.

  1. lines the authors refer to SARS-CoV however, it is not in the list of pathogenic viruses from section 2.3;

Reply: Addressed as per the reviewer’s comment in the revised manuscript.

  1. line 478, correct biopolymenrs;

Reply: Addressed as per the reviewer’s comment in the revised manuscript.

  1. line 482, correct in-vitro to in vitro;

Reply: Addressed as per the reviewer’s comment in the revised manuscript.

  1. lines 499-504, revise the text to improve language and clarity;

Reply: Addressed as per the reviewer’s comment in the revised manuscript.

  1. line 503, correct In-vivo to in vivo,;

Reply: Addressed as per the reviewer’s comment in the revised manuscript.

  1. line 504, it is not clear what is meant by “also affect our understanding of the hydrophobic residue”;

Reply: Removed this sentence in the revised manuscript.

  1. lines 505-506, include some references;

Reply: Addressed as per the reviewer’s comment in the revised manuscript.

  1. lines 537-542, the text seems unformatted;

Reply: Addressed as per the reviewer’s comment in the revised manuscript.

  1. section 4.3, line 564, indicate reference after Rapsch et al.;

Reply: Addressed as per the reviewer’s comment in the revised manuscript.

  1. line 590, E. coli should be italicized;

Reply: Addressed as per the reviewer’s comment in the revised manuscript.

  1. lines 589-592, I would suggest dividing the sentence in two;

Reply: Addressed as per the reviewer’s comment in the revised manuscript.

  1. line 602, correct and include reference to hilpert et al.;

Reply: Addressed as per the reviewer’s comment in the revised manuscript.

  1. line 604, reference to Chen et al. is missing;

Reply: Addressed as per the reviewer’s comment in the revised manuscript.

  1. line 638, include reference to Soliman et al. immediately after or at the end of the sentence;

Reply: Addressed as per the reviewer’s comment in the revised manuscript.

  1. line 659, E. coli and S. aureus should be italicized;

Reply: Addressed as per the reviewer’s comment in the revised manuscript.

  1. line 711, missing reference to Li et al.;

Reply: Addressed as per the reviewer’s comment in the revised manuscript.

  1. lines 741-742, correct in-vitro and in-vivo to in vitro and in vivo;

Reply: Addressed as per the reviewer’s comment in the revised manuscript.

  1. line 745, how was reached the conclusion that the ionic strength affects the stability of the peptide due to enzymatic degradation? The ionic strength may interfere with peptide structure stability but is doubtful that will result in enzymatic degradation.

Reply: We have revised this line after going through relevant literature. We thank the reviewer to bringing it to our attention.

Author Response

Reviewer 3:

Reviewer’s Comment The study could be organized better and the content of this review article is not descriptive enough there are few comments that need to be addressed before considering it for publication.

  1. Subtitle: 2. Peptides and Antiviral Activity
  2. The author should consider deeply discussing the potential mechanisms of action for the peptides in relation to their antiviral activity.

Reply: The mechanism of the antiviral peptides has been broadly discussed against different types of viruses in section 3.3.

  1. The author should consider discussing the potential limitations of using peptides as an antiviral therapy.

Reply: The reviewer has highlighted a very important point. We have briefly touched this topic in section 5 - Challenges with Peptide Coatings along with other limitations.

  1. The author should consider discussing the potential for peptides to used in combination with other antiviral therapies.

Reply: This is an excellent point by the reviewer. However, as we wanted to focus more on surface coating efficacy of antiviral peptides, not it’s therapeutic application, we did not include this suggestion in our revised manuscript.

  1. The author should consider discussing the potential for peptide-based therapies to be developed as prophylactics in addition to being used as treatments.

Reply: This is an excellent point by the reviewer. However, as we wanted to focus more on surface coating efficacy of antiviral peptides, not it’s therapeutic application, we did not include this suggestion in our revised manuscript.

  1. Subtitle:
  2. Current Antiviral Surface Coating Techniques
  3. It would be beneficial to include more information on the effectiveness of different antiviral surface coating techniques in reducing the spread of various viruses.
  4. A comparison of the durability and longevity of the coating techniques would also be useful in determining the overall cost-effectiveness of each method.
  5. It would be interesting to see research on the potential for these coatings to be used in everyday items, such as cell phones or public transportation surfaces. The article could also benefit from a discussion of the potential environmental impact of these coatings and any potential health risks.
  6. Potential health risks.

Reply: The reviewer has provided very useful insight to modify the manuscript here. We have captured the general role and effectiveness of current antiviral coatings in the transmission of viruses throughout the manuscript. There is no direct mention of cost-effectiveness mentioned in the papers we have reviewed. The cost-effectiveness broadly depends on the application area, the coating method,  surface preparation, and stability. We have briefly captured the potential health risk of current coatings briefly in section 5 “Challenges with Peptide Coatings”. The cost-effectiveness is currently beyond scope of our study. However, we tried to highlight the pros, cons, and durability of each current technique wherever possible.

  1. Subtitle: 4. AVPs as surface coating Discuss about:
  2. One potential criticism of using antiviral peptides as a surface coating is that the peptides may not be stable enough to remain effective over time. They may degrade or lose activity due to exposure to environmental factors such as UV light or high temperatures.
  3. Another criticism is that the peptides may not be specific enough in their targeting of viral particles, and may also harm beneficial microorganisms or cells

in the surrounding environment.

  1. It is also important to consider the cost and scalability of producing and applying the peptides as a surface coating.

Reply: These are very important points raised by the reviewer. We have captured these comments in section 5 “Challenges with Peptide Coatings”

Round 2

Reviewer 1 Report

Thank you very much for your extensive revision, it in appropriate shape for me now. 

Author Response

We are thankful to the reviewer for his/her feedback on the revised manuscript. 

We have checked for grammar and spelling mistakes and resolved all necessary writing issues.

Reviewer 2 Report

The authors have made profound changes into the manuscript that greatly improved the manuscript content and reading. Overall, the comments were adequately addressed in the revised version, even though some were not considered. Despite the great improvement on the manuscript, there are still some minor issues that need to be addressed, and are pointed below. As in the previous review, the authors should thoroughly and carefully revise the manuscript as it still contains several grammar errors. Other than that, it is my opinion that the manuscript is in a suitable form for publication.

Minor issues:

- line 157 add a space between “reactive oxygen species” and (ROS);

- line 164, suggest rephrasing to “In literature, it has been demonstrated (...)”;

- line 167, correct “damagethe”;

- lines 184 and 185, there is no need to italicize SiO2 and Ag4O4;

- lines 193-194, as mentioned in the previous review, correct the capital letter “However, Exploiting”. Also, the authors should mention what are the side effects of using metal nanoparticles. Moreover, the rationale must be improved. The authors start section 2 by mentioning that surfaces can be sterilized by using sanitizer or household cleaners, but not viable for the sterilization of surfaces after individual use. This leads the reader to the logical assumption that the authors refer to commonly used surfaces and not coatings for food purposes. In fact, in lines 195-196 it is possible to read “Incorporation of metal nanoparticles into biomaterials coatings (…)”. However, in line 194 the authors mention “They are toxic and are not meant for internal digestion”. This sentence lacks a logical flow and there is no relation of causality with the next sentence.

- subtitle of figure 6, correct “[61] [43]”;

- lines 247-250, references are missing in the description of cellulose and the other polysaccharides. Moreover, the authors should confirm that coatings based on cellulose (and the remaining polysaccharides) really display an antiviral activity (i.e. kill virus) or if they just provide a physical barrier/protection that impedes contagion;

- lines 104-105, 275-276, 299-301, refer to redundant information. The authors repeat three times the information that “peptides that have potency against viruses will be termed antiviral peptides (AVP)“. There is no need to repeat this information in lines 299-301;

- page 10, correct the large space between figure 7 and figure subtitle;

- the use of the expressions AMP and AVP should be uniform. Once the authors refer that AMPs with antiviral peptides will be termed as AVPs, the language should be uniform. For instance, in lines 304-308, the authors use AMP instead of AVP;

- in section 3.2. Human Defensin, as mentioned in the previous review, the greek letter for identification of defensins is still missing (e.g. alpha, beta). For instance, in the text, “Human defensins have only two subfamilies of cationic peptides: -defensins and -defensins”. Thoroughly revise the text of this section and make the necessary amendments.

- I would suggest to improve the final paragraph before the introduction to section 3.3. (lines 209-372);

- Review and correct the structure of subtopics from section 3.3 and onward (e.g. section 3.4. should be labelled as 3.3.1);

- Revise subtitle of figure 9 to include the complete nomenclature of the defensins;

- Page 16, correct the large space between figure 10 and figure subtitle;

- line 479, correct TiO2 to TiO2;

- line 505, either use cross-link or crosslink. Text should be uniform;

- line 537, unless there is a proved evidence of the antiviral properties of peptide-based coatings, it is suggested to change the sentence to “(…) been known to have potent antiviral properties, they will likely work effectively (…)”;

- line 578, capitalize first letter in hilpert (Hilpert);

- lines 580-582, the reference to Chen et al. is still missing;

- lines 588-589, similarly to “magainin-1 and nisin on stainless steel [129]”, provide examples of the AMPs immobilized in glass [128] and polystyrene [129];

- line 636, include the reference to Jeong et al. after the end of the sentence;

- line 721, correct “arginineglycineaspartate” to “arginine-glycine-aspartate”;

- lines 720-726, references are missing;

- page 22, correct the large space between figure 13 and figure subtitle;

- line 705, correct the protein polymer backbone used in the work by Pereira et al. to SELP-59-A.

- lines 801-804, it seems a reference is missing. 

Author Response

The authors have made profound changes into the manuscript that greatly improved the manuscript content and reading. Overall, the comments were adequately addressed in the revised version, even though some were not considered. Despite the great improvement on the manuscript, there are still some minor issues that need to be addressed, and are pointed below. As in the previous review, the authors should thoroughly and carefully revise the manuscript as it still contains several grammar errors. Other than that, it is my opinion that the manuscript is in a suitable form for publication.

We are thankful to the reviewer for his/her kind feedback.

Minor issues:

- line 157 add a space between “reactive oxygen species” and (ROS);

Addressed as per the reviewer’s comment.

- line 164, suggest rephrasing to “In literature, it has been demonstrated (...)”;

Addressed as per the reviewer’s comment.

- line 167, correct “damagethe”;

Addressed as per the reviewer’s comment.

- lines 184 and 185, there is no need to italicize SiO2 and Ag4O4;

Addressed as per the reviewer’s comment.

- lines 193-194, as mentioned in the previous review, correct the capital letter “However, Exploiting”. Also, the authors should mention what are the side effects of using metal nanoparticles. Moreover, the rationale must be improved. The authors start section 2 by mentioning that surfaces can be sterilized by using sanitizer or household cleaners, but not viable for the sterilization of surfaces after individual use. This leads the reader to the logical assumption that the authors refer to commonly used surfaces and not coatings for food purposes. In fact, in lines 195-196 it is possible to read “Incorporation of metal nanoparticles into biomaterials coatings (…)”. However, in line 194 the authors mention “They are toxic and are not meant for internal digestion”. This sentence lacks a logical flow and there is no relation of causality with the next sentence.

The paragraph is edited to address the reviewer’s comment.

- subtitle of figure 6, correct “[61] [43]”;

Addressed as per the reviewer’s comment.

- lines 247-250, references are missing in the description of cellulose and the other polysaccharides. Moreover, the authors should confirm that coatings based on cellulose (and the remaining polysaccharides) really display an antiviral activity (i.e. kill virus) or if they just provide a physical barrier/protection that impedes contagion;

Reference is added and the paragraph is edited to address reviewers’ question.

- lines 104-105, 275-276, 299-301, refer to redundant information. The authors repeat three times the information that “peptides that have potency against viruses will be termed antiviral peptides (AVP)“. There is no need to repeat this information in lines 299-301;

Addressed as per the reviewer’s comment.

- page 10, correct the large space between figure 7 and figure subtitle;

Addressed as per the reviewer’s comment.

- the use of the expressions AMP and AVP should be uniform. Once the authors refer that AMPs with antiviral peptides will be termed as AVPs, the language should be uniform. For instance, in lines 304-308, the authors use AMP instead of AVP;

Revised as per the reviewer’s comment.

- in section 3.2. Human Defensin, as mentioned in the previous review, the greek letter for identification of defensins is still missing (e.g. alpha, beta). For instance, in the text, “Human defensins have only two subfamilies of cationic peptides: -defensins and -defensins”. Thoroughly revise the text of this section and make the necessary amendments.

Revised as per the reviewer’s comment.

- I would suggest to improve the final paragraph before the introduction to section 3.3. (lines 209-372);

Revised as per the reviewer’s comment.

- Review and correct the structure of subtopics from section 3.3 and onward (e.g. section 3.4. should be labelled as 3.3.1);

Addressed as per the reviewer’s comment.

- Revise subtitle of figure 9 to include the complete nomenclature of the defensins;

Corrected as per the reviewer’s comment.

- Page 16, correct the large space between figure 10 and figure subtitle;

Addressed as per the reviewer’s comment.

- line 479, correct TiO2 to TiO2;

Addressed as per the reviewer’s comment.

- line 505, either use cross-link or crosslink. Text should be uniform;

Addressed as per the reviewer’s comment.

- line 537, unless there is a proved evidence of the antiviral properties of peptide-based coatings, it is suggested to change the sentence to “(…) been known to have potent antiviral properties, they will likely work effectively (…)”;

Addressed as per the reviewer’s comment.

- line 578, capitalize first letter in hilpert (Hilpert);

Corrected

- lines 580-582, the reference to Chen et al. is still missing;

Added reference to Chen et al.

- lines 588-589, similarly to “magainin-1 and nisin on stainless steel [129]”, provide examples of the AMPs immobilized in glass [128] and polystyrene [129];

Added 2 separate reference

- line 636, include the reference to Jeong et al. after the end of the sentence;

Addressed as per the reviewer’s comment.

- line 721, correct “arginineglycineaspartate” to “arginine-glycine-aspartate”;

Corrected

- lines 720-726, references are missing;

Added a reference

- page 22, correct the large space between figure 13 and figure subtitle;

Corrected

- line 705, correct the protein polymer backbone used in the work by Pereira et al. to SELP-59-A.

Corrected

 - lines 801-804, it seems a reference is missing

Added a reference

Reviewer 3 Report

The Authors has addressed all the questions pointed out. 

Author Response

(The authors gave the same response as above.)
